# MotifAgent: Motif-based Multi-Agent Graph-Language Alignment for Molecular Understanding and Generation

## Abstract

Large Language Models (LLMs) have shown great potential in molecular understanding by aligning molecular representations with text, enabling tasks like molecule captioning and property prediction to effectively capture molecular structures and predict functionalities. But existing approaches can only identify motifs without understanding their topological connection rules and assembly principles, preventing models from grasping the generative mechanisms of molecules. We introduce **MotifAgent**, a multi-agent reinforcement learning framework for molecular understanding. We formulate molecular assembly as a collaborative multi-agent problem, where each motif is represented by an agent sharing a common LLM backbone, dynamically reconstructing the molecule's 2D topology through global communication mechanisms. Our key innovations include: (1) inter-agent negotiation that models motif connections dynamically rather than statically; (2) a Set-based Behavioral Cloning mechanism that resolves assembly order ambiguity by learning multiple topologically equivalent paths; (3) Multi-Agent Proximal Policy Optimization (MAPPO) combined with topology-aware reward shaping to optimize target properties while maintaining chemical validity. Extensive experiments demonstrate that MotifAgent achieves substantial improvements on multiple molecule-text generation and molecular property prediction tasks, with our LLM-based generalist model surpassing or even reaching the state-of-the-art specialist models. Moreover, ablation experiments demonstrate that the MotifAgent multi-agent interaction framework can effectively learn molecular topological rules and generative principles.

## 1 Introduction

The computational representation and understanding of molecules represents a core challenge in modern drug discovery (Berdigaliyev & Aljofan, 2020) and materials design (Wang et al., 2019). With the remarkable success of large language models (LLMs) (Radford et al., 2018; Koroteev, 2021) in natural language processing, their application to molecular understanding and generation tasks (Bagal et al., 2021; Mazuz et al., 2023) has emerged as a prominent research direction in cheminformatics. Current mainstream approaches represent molecules as text sequences through SMILES strings (Mswahili & Jeong, 2024) and leverage Transformer architectures (Vaswani et al., 2017) to learn cross-modal alignment between molecular and textual representations (Zhao et al., 2023b; Song et al., 2024). These methods (Edwards et al., 2022; Zhang et al., 2024) have achieved significant progress in tasks including molecular property prediction, drug-target interaction modeling, and molecular description generation.

However, current LLM-based molecular understanding methods suffer from a fundamental limitation (Wigh et al., 2022; Bilodeau et al., 2022; Xue et al., 2019): they cannot comprehend the generative principles underlying molecular formation, particularly the connection rules between motifs and their resulting topological structures. SMILES, as a one-dimensional linear representation, must employ paired brackets and deeply nested numerical indices to encode molecular topology (Krenn et al., 2020). This linearization process inherently destroys the connectivity information present in two-dimensional molecular topology (Zhou et al., 2023). More critically, existing approaches remain limited to motif recognition while overlooking the fundamental connection rules that govern

molecular formation (Jin et al., 2020; Bettens & Lee, 2006; Collins & Bettens, 2015). The core challenge lies in understanding the chemical priors that dictate how these motifs—particularly functional groups—connect to form valid molecular structures. This involves determining which specific sites on a motif can form bonds, what types of chemical bonds are permissible (Zhang et al., 2023; Geng et al., 2023). These connection patterns directly determine molecular properties (Zhang et al., 2021). For instance, hydroxyl groups at ortho, meta, or para positions on a benzene ring exhibit different biological activities, while two aromatic rings connected through different linkers affect molecular flexibility and target binding. Existing LLM approaches treat molecules as atomic sequences or substructure collections (Zhang et al., 2024; Luo et al., 2023a; Zhao et al., 2023a), failing to understand the dynamic process by which motifs form functional wholes through specific connection rules. This prevents models from understanding motif interactions, predicting novel patterns, or generating target molecules.

To address these challenges, we propose **MotifAgent**, a **Motif-based Multi-Agent Graph-Language Alignment** framework that understands molecular assembly through generative principles. We model molecular assembly as multi-agent reinforcement learning, where LLM agents represent motifs and coordinate connection decisions through chemical constraints. We adopt a Centralized Training, Decentralized Execution (CTDE) framework with a shared LLM backbone as the policy function, enabling agents to understand their chemical environment and propose reasonable connection suggestions through negotiation. Through Set-based Behavioral Cloning (Set-BC), we avoid assembly order ambiguity by learning multiple equivalent pathways, while multi-level reward signals and potential shaping strategies guide agents to satisfy chemical validity constraints while optimizing target properties. This approach reconstructs two-dimensional topological connectivity through multi-agent global communication, where each agent maintains local connection site states and dynamically constructs inter-motif bonds. The resulting assembly graph directly encodes molecular topology, overcoming information loss inherent in linear representations. Furthermore, our framework models inter-motif connection decision processes, enabling the model to dynamically understand motif connection rules through molecular generation principles, rather than merely memorizing static patterns.

Our contributions are summarized as follows:

- To the best of our knowledge, MotifAgent is the first multi-agent framework that dynamically models molecular generative principles, moving beyond static pattern recognition to understand the connection rules governing molecular formation through collaborative agent negotiation.

- We introduce a novel approach that achieves comprehensive leading performance across molecular property prediction, description generation, and chemical reaction prediction tasks. Remarkably, MotifAgent employs general-purpose LLMs as its backbone yet achieves or surpasses existing specialized expert models, demonstrating the effectiveness of multi-agent graph-language alignment in capturing molecular semantics.

- MotifAgent provides new insights into molecular design: (1) Multi-agent collaboration naturally captures the hierarchical assembly nature of molecules while enabling controllable generation with desired functionalities through explicit connection modeling. (2) The learned motif connection rules exhibit strong generalization and chemical validity. (3) Interpretable reasoning traces reveal how motif combinations produce specific molecular properties, offering a new paradigm for understanding structure-property relationships.

## 2 RELATED WORKS

**LLM-based Molecular Understanding and Generation:** To enable large language models (LLMs) (Radford et al., 2018; Raffel et al., 2023; Touvron et al., 2023) to understand and generate molecular structures, prior works (Edwards et al., 2022; Christofidellis et al., 2023; Liu et al., 2023b; Li et al., 2024; Zhang et al., 2024) jointly train on molecular SMILES string representations and natural language text, allowing models to learn bidirectional conversions between molecular structures and textual descriptions. For instance, molT5 (Edwards et al., 2022) learns SMILES-to-text translation in a self-supervised manner and first achieved success on text-attributed tasks. Meanwhile, approaches like (Su et al., 2022; Liu et al., 2023a; Luo et al., 2023a; Liu et al., 2023c; Zhao et al., 2023a) incorporate 2D molecular graph information and employ multimodal contrastive learning to unify the representations of molecular graphs, SMILES strings, and textual descriptions through global

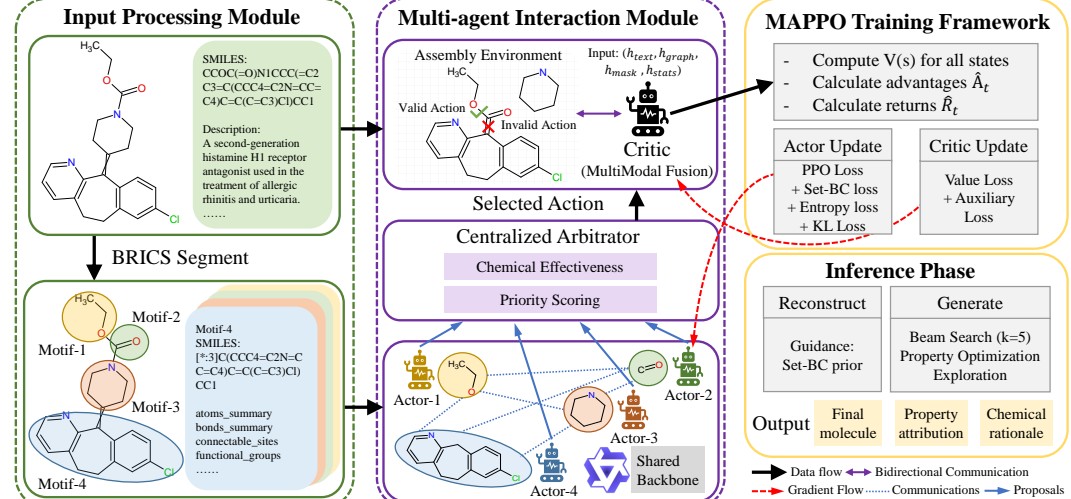

Figure 1: Overview of the proposed MotifAgent framework. MotifAgent consists of four integrated components: (1) Input Processing Module decomposes molecules into motifs using BRICS segmentation and converts them to structured text representations. (2) Multi-agent Interaction Module employs LLM-based actors (one per motif) sharing a common backbone to propose connections, with a Centralized Arbitrator selecting valid actions and a Critic evaluating assembly states through multi-modal fusion. (3) MAPPO Training Framework jointly optimizes Actor and Critic networks with separate loss functions. (4) Inference Phase supports both reconstruction mode with Set-BC guidance and generation mode for property optimization.

alignment. Our method advances this research by introducing multiple LLM agents to model connectivity among molecular motifs, enabling LLMs to understand molecular generation through substructure assembly.

**Multi-Agent Reinforcement Learning:** MARL (Canese et al., 2021; Wen et al., 2022; Albrecht et al., 2024) has emerged as a central paradigm for coordinating multiple autonomous agents in shared environments. Compared to single large models, multi-agent systems (MAS) offer enhanced scalability and task decomposition through role-specific agents (Gao et al., 2025), particularly valuable for molecular design, where specialized agents can focus on different chemical motifs. The Centralized Training with Decentralized Execution (CTDE) framework (Lowe et al., 2017; Sunehag et al., 2017; Rashid et al., 2020) effectively addresses fundamental challenges such as policy non-stationarity and partial observability by leveraging global information during training while maintaining decentralized policies during execution. Methods like MAPPO (Lohse et al., 2021) use shared policy networks where agents make decisions based on local observations. Our method employs this CTDE framework with a shared LLM backbone, enabling motif agents to learn global molecular topology during training while making decentralized connection decisions based on local chemical environments during execution.

## 3 METHOD

We formulate fragment-based molecular assembly as a centralized training with decentralized execution (CTDE) multi-agent reinforcement learning problem. The core idea is to employ a shared large language model (LLM) as the decentralized policy backbone, enabling each motif agent to propose connections under textual chemical constraints while dynamically reconstructing the molecule's 2D topology through global communication. A centralized critic receives global graph information and mask features during training to evaluate state values and assembly progress. We adopt Multi-Agent Proximal Policy Optimization (MAPPO) for stable and efficient policy updates. To balance reconstruction (strictly restoring the original molecule) and generation (optimizing properties under chemical validity) objectives, we introduce Set-based Behavior Cloning (Set-BC) to avoid assembly order ambiguity, potential shaping to accelerate target graph alignment, and connectivity/over-connection constraints to regularize the topological process.

Figure 2: Hierarchical Sampling Process of MotifAgent.

## 3.1 REPRESENTATION AND FRAGMENTATION

We decompose molecules into chemically meaningful bricks (containing at least 2 non-hydrogen atoms) and linkers following an improved BRICS algorithm with 16 chemical environment-specific bond-breaking rules, while preserving complete metadata including connection sites and allowed bond types. To facilitate LLM processing and reconstruct 2D topological information, we serialize each motif and its connectivity into structured text. Each motif description contains a unique identifier, SMILES string, atom and bond information, connectable sites (including site type, chemical environment, allowed bond types), and property summaries such as aromaticity, ring structures, and functional groups. Connections are represented using a unified template: `CONNECTION: motif_i[site_x] --bond_type--> motif_j[site_y]`, which directly encodes adjacency relationships between motifs and preserves the molecular graph's connectivity information, ensuring the correct topological structure is maintained.

## 3.2 ENVIRONMENT MODELING

We model the assembly process as a Dec-POMDP where the state space directly corresponds to the molecule's 2D topological evolution. The global state at time $t$ consists of the current assembly graph $G_t = (V, E_t)$'s textual summary, unconnected motif list, available site topology (which motifs can interconnect), and optionally online property estimates, where $V$ represents the motif node set and $E_t$ denotes the edge set (established connections) at time $t$. The centralized critic additionally receives structured graph features and mask tensors during training to perceive global topology. Actions are defined as connections between two motifs' specified sites with a particular bond type: $a_t = (i, s_i, j, s_j, b)$, where $i, j$ are motif indices, $s_i, s_j$ are connection sites, and $b$ is the bond type. Each action directly modifies the molecule's 2D topological structure, with an explicit STOP action to indicate termination. To reduce action branching while maintaining topological consistency, we employ hierarchical sampling: first selecting source motif/site, then selecting target motif/site based on chemical compatibility and topological constraints, finally selecting bond type. Each layer applies chemical and topological masks at the logit level to eliminate invalid options, ensuring generated connections maintain 2D topological validity. After receiving an action, the environment performs valence, aromaticity, and topological feasibility checks (e.g., avoiding unreasonable small rings), updates the assembly graph's adjacency matrix and available sites, and outputs rewards and the next topological state.

**Precise termination conditions**: In reconstruction mode, necessary conditions for termination are $cc(G_t) = 1$ and $E_t \supseteq E^*$, where $cc(\cdot)$ denotes the number of connected components and $E^*$ is the target molecule's edge set. Sufficient conditions are $E_t = E^*$ or the policy selecting STOP, with timeout protection (steps $> 2|E^*|$) to prevent infinite loops. In generation mode, termination conditions include: (1) chemical completeness—all required valences saturated; (2) topological completeness—forming a connected molecular graph; (3) property convergence—improvement over $k$ consecutive steps $< \varepsilon$, where $k$ is the window size and $\varepsilon$ is the convergence threshold; (4) active termination—policy outputting STOP probability $> 0.9$.

## 3.3 AGENTS AND CRITIC

Each motif corresponds to an agent, with all agents sharing the same LLM as the policy backbone, generating connection proposals and chemical rationales conditioned on their motif descriptions and the current global assembly's topological summary. Crucially, each agent not only knows its local structure but also perceives the molecule's 2D topological state through global communication.

**LLM policy network explicitly models connection decisions**: During hierarchical action sampling, the LLM generates action distributions through specific prompt templates, with each layer considering current topological constraints. When selecting the source motif in the first layer, the LLM evaluates each available motif's connection potential within the current topology. The second layer, conditioned on the selected motif, evaluates each site's chemical activity and topological accessibility. The third layer evaluates target compatibility based on the source's chemical environment and global topology. The fourth layer determines optimal bond type based on both ends' chemical environments and target topological structure. The implementation uses the LLM's last hidden layer features to generate logits through trainable action heads:

$$\pi(a|s) = \text{Softmax}(\text{MLP}(\text{LLM}_{\text{hidden}}(\text{prompt}(s)))) \tag{1}$$

where $\pi$ is the policy function, $a$ is the action, $s$ is the state, $\text{LLM}_{\text{hidden}}$ denotes the LLM's hidden representation, and $\text{prompt}(s)$ is the prompt text containing the complete description of the current 2D topology.

**Central arbitrator coordinates topological construction**: The arbitrator employs two-phase coordination to ensure topological consistency. Phase 1 performs chemical and topological validity screening: collecting all motif agents' proposals in parallel, quickly filtering out proposals violating valence rules or destroying topological integrity through a rule engine. Phase 2 conducts topological priority scoring and selection with the scoring function:

$$S(a) = w_1 \cdot \text{ChemStability}(a) + w_2 \cdot \text{TopoProgress}(a) + w_3 \cdot \text{PropImprove}(a) \tag{2}$$

where $S(a)$ is action $a$'s score, $w_1, w_2, w_3$ are weight coefficients, and topological progress explicitly considers contributions to 2D topological integrity such as reducing connected components, adding key bridging connections, and forming stable ring systems.

**Fused representation perceives global topology**: The centralized critic $V_\phi(x_t)$'s fused representation integrates topological information through multi-modal attention:

$$x_t = \text{MultiModalFusion}([h_{\text{text}}, h_{\text{graph}}, h_{\text{mask}}, h_{\text{topo}}]) \tag{3}$$

where $V_\phi$ is the value function, $\phi$ denotes critic network parameters, $x_t$ is the fused representation at time $t$, $h_{\text{text}}$ is the LLM-encoded global state text representation, $h_{\text{graph}}$ is the GNN-encoded current assembly graph's 2D topological structure, $h_{\text{mask}}$ is the available action mask based on topological constraints, and $h_{\text{topo}}$ is the MLP-encoded topological statistics (connected components, ring count, shortest path distribution, etc.).

To enhance perception of assembly progress and topological evolution, we attach multi-task regression heads beyond the value head to predict remaining target edges and connected components, forming value signals consistent with topological progress.

## 3.4 Rewards and Shaping

The reward design explicitly considers 2D topology reconstruction and optimization. Chemical base rewards include validity $r_{\text{valid}}$ (+1 for valid, -1 for invalid), local stability $r_{\text{stable}} = -E_{\text{strain}}$ where $E_{\text{strain}}$ is force field-calculated strain energy, functional group formation $r_{\text{func}} = \sum_g w_g \mathbb{I}\{\text{form } g\}$ where $g$ iterates over all functional group types with weights $w_g$ and indicator function $\mathbb{I}$, target property alignment $r_{\text{prop}} = -|y_{\text{pred}} - y_{\text{target}}|$ where $y_{\text{pred}}$ and $y_{\text{target}}$ are predicted and target property values respectively, synthetic accessibility $r_{\text{SA}}$, and novelty $r_{\text{novel}} = 1 - \max_{\text{ref}} \text{Tanimoto}(\text{mol}, \text{ref})$ where Tanimoto measures molecular similarity.

For topological shaping, we design specific rewards to guide correct 2D topology construction. Cross-component connection reward accelerates connected graph formation through $r_{\text{conn}}(s_t, s_{t+1}) = (\text{cc}(G_t) - 1) - (\text{cc}(G_{t+1}) - 1)$ where $s_t$ and $s_{t+1}$ are adjacent states, yielding positive reward when reducing connected components. Target edge progression reward ensures accurate target topology recovery via $r_{\text{edge}}(s_t, s_{t+1}) = |E^* \cap E_{t+1}| - |E^* \cap E_t|$ where $\cap$ denotes set intersection, equaling the number of newly added correct edges. Topological distance reward encourages formation of target-similar topological features using $r_{\text{topo}}(s_t, s_{t+1}) = -\text{GraphEditDistance}(G_{t+1}, G^*)$. To prevent topological structure destruction, we penalize connections exceeding the target edge count during reconstruction with $r_{\text{over}}(s_{t+1}) = -\alpha \cdot \max(0, |E_{t+1}| - |E^*|)$ where $\alpha$ is the penalty coefficient.

Potential shaping is designed based on topological differences with potential function $\Phi(s) = -|E^* \setminus E(s)| - \beta \cdot \text{cc}(G_s)$ where $\setminus$ denotes set difference and $\beta$ is the connectivity weight. The

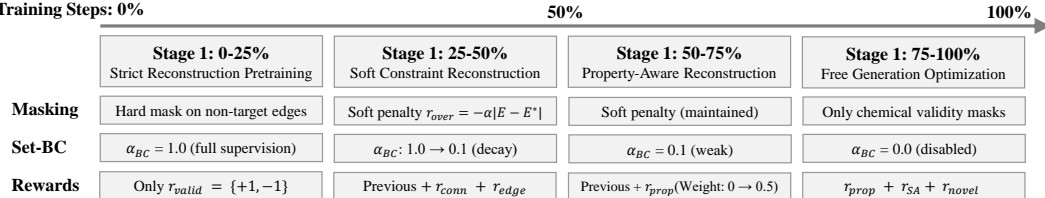

Figure 3: Curriculum Learning Training for MotifAgent.

shaped reward becomes $r'(s_t, a_t, s_{t+1}) = r(s_t, a_t, s_{t+1}) + \gamma\Phi(s_{t+1}) - \Phi(s_t)$ where $\gamma$ is the discount factor. The combined single-step reward is:

$$R = \underbrace{r_{\text{valid}} + r_{\text{stable}} + r_{\text{func}} + r_{\text{prop}} + r_{\text{SA}} + r_{\text{novel}}}_{\text{Chemical base rewards}} + \underbrace{r_{\text{conn}} + r_{\text{edge}} + r_{\text{topo}}}_{\text{Topological shaping rewards}} \tag{4}$$

## 3.5 POLICY OPTIMIZATION AND TRAINING

We employ MAPPO (Lohse et al., 2021) for policy updates, crucially enabling the policy to learn motif connection rules rather than memorizing specific sequences. During training, the Actor (policy network) and Critic are optimized separately using different loss functions.

**Actor loss function**: The policy network is optimized through:

$$\mathcal{L}_{\text{actor}} = -\mathcal{L}_{\text{clip}} - \beta\mathcal{H}(\pi_\theta) + \alpha_{\text{BC}}\mathcal{L}_{\text{BC}} + \mathcal{L}_{\text{KL}} - \sum_k \lambda_k \mathbb{E}[c_k(s)] \tag{5}$$

where $\mathcal{L}_{\text{clip}}$ is the PPO clipped objective optimizing policy to maximize expected reward, $\beta\mathcal{H}(\pi_\theta)$ provides entropy regularization for exploration, $\alpha_{\text{BC}}\mathcal{L}_{\text{BC}}$ is the Set-BC supervision term learning correct assembly patterns, $\mathcal{L}_{\text{KL}}$ constrains policy change relative to reference policy, and $\sum_k \lambda_k \mathbb{E}[c_k(s)]$ represents constraint terms satisfying chemical and topological requirements.

The PPO clipped objective takes the form:

$$\mathcal{L}_{\text{clip}}(\theta) = \mathbb{E}_{i,t}\left[\min\left(r_t^i(\theta)\hat{A}_t, \text{clip}(r_t^i(\theta), 1-\epsilon, 1+\epsilon)\hat{A}_t\right)\right] \tag{6}$$

where $r_t^i(\theta) = \pi_\theta(a_t^i|o_t^i)/\pi_{\theta_{\text{old}}}(a_t^i|o_t^i)$ is the importance sampling ratio, and $\hat{A}_t$ is the advantage estimate computed using Generalized Advantage Estimation (GAE).

**Critic loss function**: The critic network is optimized independently with:

$$\mathcal{L}_{\text{critic}} = \mathbb{E}_t[(V_\phi(x_t) - \hat{R}_t)^2] + w_1(V_{\text{edges}}(x_t) - |E^* \setminus E_t|)^2 + w_2(V_{\text{cc}}(x_t) - \text{cc}(G_t))^2 \tag{7}$$

The main value loss $(V_\phi(x_t) - \hat{R}_t)^2$ uses Monte Carlo return estimates, while auxiliary heads $V_{\text{edges}}$ and $V_{\text{cc}}$ predict remaining target edges and connected components respectively with weights $w_1, w_2$.

**Set-BC learns topologically equivalent paths**: To address multiple assembly sequences reaching the same topology, Set-BC avoids enforcing specific orders by maximizing policy probability over the entire correct action set $\mathcal{A}_t^*$:

$$\mathcal{L}_{\text{BC}} = -\mathbb{E}_t\left[\log \sum_{a \in \mathcal{A}_t^*} \pi_\theta(a|s_t)\right] \tag{8}$$

where $\mathcal{A}_t^*$ contains all actions preserving topological correctness at state $s_t$, $\pi_\theta(a|s_t)$ is the policy probability for action $a$, and the expectation is over time steps $t$.

**Curriculum learning progressively masters topological construction**: Training proceeds through four phases, transitioning from simple topological constraints to complex optimization. Phase 1 (first 25%) focuses on strict topological reconstruction with hard masks and Set-BC weight $\alpha_{\text{BC}} = 1.0$. Phase 2 (25%-50%) introduces soft-constraint learning with $\alpha_{\text{BC}}$ decaying to 0.1. Phase 3 (50%-75%) combines topology-aware property optimization. Phase 4 (final 25%) enables free topological exploration by removing hard constraints.

**Computational efficiency optimization** focuses on key topological operations through topology caching of frequent operations, incremental topology updates for local changes, and topology batching for similar states. The training pipeline samples diverse construction paths through parallel environments while maintaining topological consistency and jointly optimizing all loss components.

## 3.6 INFERENCE AND DOWNSTREAM APPLICATIONS

During inference, the policy assembles structures based on learned topological rules, with hierarchical masks ensuring topological validity. Reconstruction tasks succeed when the target topology is fully recovered ($E = E^*$ and $\text{cc}(G) = 1$). Generation tasks optimize properties within topological constraints after forming a connected topology, exploring multiple paths via beam search (beam width $k = 5$).

To improve efficiency, we use: (1) topological pruning—early elimination of infeasible paths based on learned patterns; (2) topology-guided sampling—prioritizing actions that quickly form stable topologies; (3) topological checkpoints—saving key states for backtracking and branching.

For downstream tasks, the system provides three explanation levels: (1) topological construction—how each step changes the molecule's connectivity; (2) motif connection mechanisms—chemical reasons for bond choices; (3) topology-property relationships—topological features influencing properties. By reusing LLM encodings, property prediction heads leverage complete 2D topological information, achieving excellent performance.

This system not only reconstructs molecules' 2D topologies, overcoming linear representation's limitations, but also learns motif connection rules and molecular assembly principles through multi-agent collaboration, providing a topology-aware tool for designing functional molecules.

## 4 EXPERIMENTS

### 4.1 INITIAL TRAINING

**Datasets and Training Details**: We utilize the same molecular SMILES-text pairs dataset as MoleculeSTM Liu et al. (2023a), collected from the PubChem website. Following their preprocessing pipeline, pairs with identical PubChem IDs and descriptions shorter than 18 characters are merged, with duplicates removed from downstream task datasets to prevent data leakage. This yields 51,340 unique high-quality pairs for initial training. For each molecule, we apply our improved BRICS fragmentation to generate motif sets, with molecules containing 2-15 motifs selected for training. The model employs Qwen2.5-7B as the shared policy backbone and MolT5-base as the centralized critic. All other training configurations and hyperparameters are detailed in Appendix A.

### 4.2 MOLECULAR DESCRIPTION GENERATION

To evaluate MotifAgent's performance on molecular description generation, we adopt the widely-used ChEBI-20 benchmark dataset (Papadatos et al., 2015), which requires models to generate natural language descriptions of chemical features, properties, and biological activities given molecular structures. Following standard protocols, we employ BLEU, ROUGE, and METEOR metrics to assess the quality of generated text.

Experimental results on Table 1 demonstrate that MotifAgent achieves state-of-the-art performance among LLM-based generalist models, substantially outperforming existing general-purpose methods across all metrics. Compared to the second-best HIGHT-GS (Chen et al., 2025), MotifAgent delivers an average performance improvement of 22.5%. This improvement demonstrates effective motif connection modeling via multi-agent collaboration. More notably, MotifAgent exhibits strong competitiveness against specialist models (Edwards et al., 2022; Su et al., 2022; Luo et al., 2023a; Liu et al., 2023b; 2024; 2023c; Christofidellis et al., 2023; Zhang et al., 2024) specifically designed for molecule-text tasks, surpassing all specialist baselines on the majority of metrics. This indicates that explicit learning of molecular topology and motif connection rules effectively compensates for the lack of domain-specific pretraining in generalist models (Luo et al., 2023b; Cao et al., 2023; Fang et al., 2023; Chen et al., 2025). Ablation analysis comparing MotifAgent with and without initial training reveals consistent improvements across all metrics in the complete version, validating the importance of our curriculum learning strategy: first establishing fundamental topological reconstruction capabilities, then optimizing for task-specific objectives, thereby building a more robust foundation for molecular understanding. Collectively, these results demonstrate that through multi-agent collaboration and explicit connection rule learning, MotifAgent genuinely comprehends molecular assembly principles, enabling generation of more accurate chemical descriptions.

Table 1: Performance comparison on molecular description generation task. The top **1st** and 2nd results are highlighted.

| Method | BLEU-2↑ | BLEU-4↑ | ROUGE-1↑ | ROUGE-2↑ | ROUGE-L↑ | METEOR↑ |
|---|---|---|---|---|---|---|
| *Specialist Models* | | | | | | |
| MoT5-base (Edwards et al., 2022) | 0.540 | 0.457 | 0.634 | 0.485 | 0.568 | 0.569 |
| MoMu (MolT5-base) (Su et al., 2022) | 0.549 | 0.462 | - | - | - | 0.576 |
| MolFM (MolT5-base) (Luo et al., 2023a) | 0.585 | 0.498 | 0.653 | 0.508 | 0.594 | 0.607 |
| MolXPT (Liu et al., 2023b) | 0.594 | 0.505 | 0.660 | 0.511 | 0.597 | 0.626 |
| GIT-Mol-(graph+SMILES) (Liu et al., 2024) | 0.352 | 0.263 | 0.575 | 0.485 | 0.560 | 0.430 |
| MolCA, Galac1.3B (Liu et al., 2023c) | 0.620 | 0.531 | 0.681 | 0.537 | 0.618 | - |
| Text+Chem T5-augm-base (Christofidellis et al., 2023) | 0.625 | 0.542 | 0.682 | 0.543 | 0.622 | 0.648 |
| Atomas-base (Zhang et al., 2024) | 0.632 | **0.549** | 0.685 | 0.545 | 0.626 | - |
| *Retrieval Based LLMs* | | | | | | |
| GPT-4-0314 (10-shot MolReGPT) (Li et al., 2024) | 0.607 | 0.525 | 0.634 | 0.476 | 0.562 | 0.610 |
| *LLM Based Generalist Models* | | | | | | |
| BioMedGPT-10B (Luo et al., 2023b) | 0.234 | 0.141 | 0.386 | 0.206 | 0.332 | 0.308 |
| InstructMol-GS (Cao et al., 2023) | 0.453 | 0.349 | 0.546 | 0.372 | 0.482 | 0.483 |
| Mol-Instruction (Fang et al., 2023) | 0.249 | 0.171 | 0.331 | 0.203 | 0.289 | 0.271 |
| HIGHT-GS (Chen et al., 2025) | 0.498 | 0.397 | 0.582 | 0.414 | 0.518 | 0.525 |
| MotifAgent w/o initial training | 0.617 | 0.509 | 0.641 | 0.523 | 0.582 | 0.620 |
| MotifAgent | **0.642** | 0.545 | **0.686** | **0.557** | **0.633** | **0.651** |

Table 2: Performance comparison on retrosynthesis prediction tasks.

| Method | EXACT↑ | BLEU↑ | LEVENSHTEIN↓ | RDK FTS↑ | MACCS FTS↑ | MORGAN FTS↑ | VALIDITY↑ |
|---|---|---|---|---|---|---|---|
| *Retrosynthesis* | | | | | | | |
| Alpaca (Dubois et al., 2023) | 0.000 | 0.063 | 46.915 | 0.005 | 0.023 | 0.007 | 0.160 |
| Baize (Xu et al., 2023) | 0.000 | 0.095 | 44.714 | 0.025 | 0.050 | 0.023 | 0.112 |
| ChatGLM (Zeng et al., 2022) | 0.000 | 0.117 | 48.365 | 0.056 | 0.075 | 0.043 | 0.046 |
| LLama (Touvron et al., 2023) | 0.000 | 0.036 | 46.844 | 0.018 | 0.029 | 0.017 | 0.010 |
| Vicuna (Chiang et al., 2023) | 0.000 | 0.057 | 46.877 | 0.025 | 0.030 | 0.021 | 0.017 |
| Mol-Instruction (Fang et al., 2023) | 0.009 | 0.705 | 31.227 | 0.283 | 0.487 | 0.230 | **1.000** |
| Llama-7b (Touvron et al., 2023)(LoRA) | 0.000 | 0.283 | 53.510 | 0.136 | 0.294 | 0.106 | **1.000** |
| InstructMol-GS (Cao et al., 2023) | 0.172 | 0.911 | 20.300 | 0.765 | 0.615 | 0.568 | **1.000** |
| HIGHT-GS (Chen et al., 2025) | 0.202 | 0.914 | 20.194 | 0.772 | 0.623 | 0.577 | 0.999 |
| MotifAgent | **0.275** | **0.932** | **18.810** | **0.783** | **0.685** | **0.631** | **1.000** |

## 4.3 CHEMICAL REACTION PREDICTION

To evaluate MotifAgent's capability in chemical reaction prediction tasks, we conduct comprehensive experiments on the Mol-Instructions dataset. We present only the most challenging retrosynthesis prediction task here, with complete results for reagent prediction and forward reaction prediction available in Appendix D. Retrosynthesis prediction, which requires models to infer suitable reactants given target products, represents a fundamental challenge in AI-assisted synthetic route planning. We employ both linguistic distance metrics (BLEU, Levenshtein) and molecular fingerprint similarities (RDK, MACCS, MORGAN FTS) for comprehensive evaluation.

Table 2 presents the retrosynthesis prediction results, where MotifAgent outperforms all baseline methods across key metrics. It demonstrates superior exact match accuracy, sequence generation quality, and molecular structure similarity. MotifAgent excels in modeling molecular formation through effective multi-agent collaboration, identifying and tracking structural transformations at the motif level, which is essential for recognizing reaction centers. Its 100% chemical validity further highlights the structural integrity ensured by our approach. These results establish motif-based multi-agent coordination as a powerful tool for AI-driven retrosynthetic analysis.

## 4.4 MOLECULAR PROPERTY PREDICTION

We evaluate MotifAgent on 8 benchmark datasets from MoleculeNet (Wu et al., 2018) for molecular property classification, using ROC-AUC as the evaluation metric. Following prior work, all datasets are converted to instruction format. Molecules are decomposed into motif sets via the BRICS algorithm with corresponding textual descriptions constructed.

Table 3 shows MotifAgent's superior performance, with an average ROC-AUC of 77.19, significantly outperforming existing LLM-based models like InstructMol and HIGHT. It achieves a 24.4% improvement over its backbone model, Qwen2.5-7B, validating the effectiveness of our multi-agent framework in reconstructing 2D molecular structures and learning motif connection rules. MotifAgent excels as a generalist framework, surpassing specialist models on tasks like Tox21, HIV, and Bace, bridging the gap between generalist and specialist models. This demonstrates the potential of LLM-based molecular-text representation frameworks to achieve both generality and specialization, offering valuable insights for future research.

Table 3: Performance comparison on molecular classification tasks.

| Method | BBBP | Tox21 | ToxCast | Sider | ClinTox | MUV | HIV | Bace | Avg |
|---|---|---|---|---|---|---|---|---|---|
| *Specialist Models* | | | | | | | | | |
| MoleculeSTM-SMILES (Liu et al., 2023a) | 70.75±1.9 | 75.7±0.9 | 65.3±0.37 | 63.7±0.81 | 86.6±2.28 | 65.7±1.46 | 77.0±0.4 | 81.9±0.4 | 73.33 |
| MolFM (Luo et al., 2023a) | 72.9±0.1 | 77.2±0.7 | 64.4±0.2 | 64.2±0.9 | 79.7±1.6 | 76.0±0.8 | 78.8±1.1 | 83.9±1.1 | 74.62 |
| MoMu (Su et al., 2022) | 70.5±2.0 | 75.6±0.3 | 63.4±0.5 | 60.5±0.9 | 79.9±4.1 | 70.5±1.4 | 75.9±0.8 | 76.7±2.1 | 71.63 |
| MolCA-SMILES (Liu et al., 2023c) | 70.8±0.6 | 76.0±0.5 | 56.2±0.7 | 61.1±1.2 | 89.0±1.7 | - | - | 79.3±0.8 | 72.1 |
| Atomas (Zhang et al., 2024) | **73.7±1.7** | 77.8±0.4 | 66.9±0.9 | 64.4±1.9 | **93.1±0.5** | 76.3±0.7 | 80.5±0.43 | 83.1±1.7 | 77.01 |
| *LLM Based Generalist Models* | | | | | | | | | |
| Qwen2.5-7b (Hui et al., 2024) | 59.7±0.7 | 62.7±0.5 | 57.3±1.1 | 52.9±0.9 | 71.0±1.8 | 60.9±01.5 | 61.1±0.9 | 70.3±0.8 | 62.05 |
| InstructMol (Cao et al., 2023) | 55.4 | - | - | - | - | - | 57.5 | 63.2 | 58.70 |
| HIGHT (Chen et al., 2025) | 59.4 | - | - | - | - | - | 58.6 | 68.4 | 62.13 |
| MotifAgent | 73.4±0.8 | **78.5±0.4** | **67.6±0.8** | **65.1±1.3** | 90.9±0.7 | **77.4±0.6** | **80.6±0.4** | **84.0±1.2** | **77.19** |

Table 4: Ablation study for the effectiveness of Multi-Agent Collaboration.

| | Reconstruction Accuracy (%) | Chemical Validity (%) | Graph Edit Distance ↓ | Morgan FTS ↑ | Connection Site Accuracy (%) | Bond Type Accuracy (%) |
|---|---|---|---|---|---|---|
| Single-Agent | 67.1 | 82.3 | 5.85 | 0.708 | 68.4 | 76.9 |
| Multi-Agent (Ours) | 85.7 | 95.6 | 4.31 | 0.792 | 87.6 | 92.3 |

## 4.5 ABLATION STUDIES

To validate the key design choices in our MotifAgent framework, we conduct comprehensive ablation studies focusing on two critical components: (1) the multi-agent collaboration mechanism for understanding motif connection rules and reconstructing 2D topology, and (2) the Set-based Behavior Cloning (Set-BC) for learning topologically equivalent assembly paths. Details of the metrics used in the experiments can be found in the appendix D.

**Multi-Agent Collaboration vs. Single Sequential Generation.** To validate the necessity of multi-agent collaboration for learning motif connection rules, we compare our full MotifAgent against a single-agent baseline that concatenates all motifs into one sequence for sequential connection decisions. Both models use identical LLM backbones (Qwen2.5-7B), reward functions, and hyperparameters. As shown in Table 4, the multi-agent system significantly outperforms the single-agent baseline across all metrics. The superior performance in Graph Edit Distance and Morgan fingerprint similarity demonstrates that multi-agent collaboration better preserves 2D topological structure, while the improved connection-level accuracies reveal that distributed agents with specialized local knowledge make more informed decisions about connection sites and bond types than a single agent managing all motifs simultaneously.

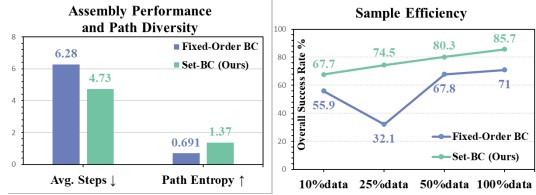

Figure 4: Ablation study for the effectiveness of the Set-BC.

**Set-BC vs. Fixed-Order Supervision.** The second ablation examines Set-BC's importance in handling molecular assembly sequence ambiguity. Since molecules can be correctly assembled through multiple equivalent paths, forcing models to learn single arbitrary sequences may hinder learning efficiency and generalization. We compare Set-BC against a fixed-order baseline using traditional behavior cloning with BFS-determined assembly sequences. As shown in Figure 4, the results demonstrate the superiority of Set-BC over fixed-order supervision across all metrics. Set-BC achieves more efficient assembly with fewer average steps and exhibits higher path entropy, confirming successful learning of diverse assembly strategies. Most significantly, Set-BC maintains superior sample efficiency across all data scales, demonstrating that avoiding assembly order ambiguity is crucial for both learning efficiency and generalization.

## 5 CONCLUSION

We presented MotifAgent, a multi-agent reinforcement learning framework that effectively addresses LLMs' limitations in understanding molecular generation principles. Our approach explicitly learns motif connection rules governing molecular topology, and leverages the CTDE framework combined with Set-BC to learn from multiple equivalent assembly paths. Experiments demonstrate that MotifAgent achieves state-of-the-art performance on molecular property prediction, description generation, and chemical reaction prediction tasks, proving its generalization and scalability.

## 6 ETHICS STATEMENT

All data used in this study are publicly available and do not contain personally identifiable information. The research was conducted in accordance with the ethical guidelines for computational research, ensuring that all methodologies and procedures followed appropriate ethical standards. The authors declare that they have no conflicts of interest related to this study.

## 7 REPRODUCIBILITY STATEMENT

All experimental code will be made publicly available upon paper acceptance, while detailed hyper-parameters, experimental configurations, and hardware setup are provided in Appendix A.

## 8 LLM USAGE

Large language models (LLMs) were used for refining sentence structure, improving grammatical accuracy, and enhancing the clarity of the manuscript text. A supporting role was played by the LLMs in the manuscript's language polishing, but no scientific content, data analysis, or experimental design was generated by the LLMs.

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

## A  INITIAL TRAINING DETAILS

We provide comprehensive training configuration and hyperparameter settings for MotifAgent in Table 5. The model is trained on ith 8 NVIDIA Tesla A100 GPUs (80GB RAM/GPU) with Qwen2.5-7B as the shared policy backbone and MolT5-base as the centralized critic. We employ 32 parallel environments with a rollout length of 64 steps, resulting in 2,048 transitions per update. The actor learning rate is set to 5e-5 with LoRA (rank=16), while the critic uses 3e-4. The training employs curriculum learning with four phases, automatically transitioning based on performance metrics to progressively master topological construction from strict reconstruction to free exploration. For curriculum learning, Set-BC weight $\alpha_{BC}$ decays from 1.0 to 0.1 over the first 50

**Curriculum Learning Details**: The four-phase curriculum is designed to progressively build the model's capabilities: Phase 1 (0-25%): Strict reconstruction with hard masks, focusing on learning valid chemical connections; Phase 2 (25-50%): Soft constraints with Set-BC weight decay, allowing exploration of equivalent paths; Phase 3 (50-75%): Property-aware reconstruction, introducing target property rewards; Phase 4 (75-100%): Free exploration for generation, removing hard topological constraints. The automatic phase transitions ensure the model has sufficiently mastered each level before progressing, preventing premature exploration that could lead to unstable training.

## B  CHEMICAL REACTION PREDICTION

We evaluated MotifAgent on three chemical reaction prediction tasks from the Mol-Instructions dataset Fang et al. (2023): reagent prediction, forward reaction prediction, and retrosynthesis. These tasks are crucial for AI-assisted drug discovery. All inputs and outputs adopt SELFIES representation. Evaluation metrics include linguistic distance measures (BLEU, Levenshtein distance) and molecular fingerprint similarities (RDK FTS, MACCS FTS, MORGAN FTS) computed via RDKit.

Table 5: Hyperparameter details for MotifAgent.

| Hyperparameter | Value |
|---|---|
| *Model Architecture* | |
| policy backbone | Qwen2.5-7B |
| critic network | MolT5-base |
| LoRA rank | 16 |
| action head hidden dim | 512 |
| *Training Configuration* | |
| parallel environments | 8 |
| rollout length | 32 |
| batch size (transitions) | 256 |
| PPO epochs | 10 |
| max training steps | 200K |
| max sequence length | 512 |
| precision | BFloat16 Automatic Mixed Precision |
| *Optimization* | |
| actor learning rate | 1e-5 |
| critic learning rate | 1e-4 |
| gradient clip norm | 0.5 |
| optimizer | AdamW |
| warmup steps | 10000 |
| *PPO Parameters* | |
| clipping $\epsilon$ | 0.2 |
| GAE $\lambda$ | 0.95 |
| discount $\gamma$ | 0.99 |
| value loss coefficient | 0.5 |
| *Regularization* | |
| entropy $\beta$ (initial) | 0.01 |
| entropy $\beta$ (final) | 0.001 |
| KL penalty $\beta_{KL}$ | 0.1 |
| Set-BC $\alpha_{BC}$ (initial) | 1.0 |
| Set-BC $\alpha_{BC}$ (final) | 0.1 |
| Set-BC decay steps | 50% of training |
| *Auxiliary Tasks* | |
| edge prediction weight $w_1$ | 0.1 |
| component prediction weight $w_2$ | 0.1 |

Table 5 shows that MotifAgent achieves state-of-the-art performance across all three tasks. For reagent prediction, MotifAgent attains 8.5% exact match rate (26.9% improvement over HIGHT-GS), 0.516 BLEU score, and 22.571 Levenshtein distance, outperforming all baselines including Mol-Instruction which uses Llama-2 Touvron et al. (2023) backbone. In forward reaction prediction, MotifAgent achieves 31.5% exact match rate and 0.937 BLEU score, with molecular fingerprint similarities reaching 0.806 (RDK FTS), 0.669 (MACCS FTS), and 0.582 (MORGAN FTS), all setting new records. For the most challenging retrosynthesis task, MotifAgent reaches 27.5% exact match rate (36.1% relative improvement over HIGHT-GS), with MORGAN FTS achieving 0.631, significantly higher than other methods.

MotifAgent's superior performance stems from its multi-agent collaborative mechanism that understands chemical reactions at the motif level—modeling functional group transformations, reaction center identification, and electron transfer paths. Each motif agent encodes local chemical environments while perceiving global reaction changes through communication. The Set-BC mechanism enables learning multiple equivalent reaction pathways, crucial for reactions with multiple mechanisms. All tasks achieve 100% chemical validity, demonstrating the effectiveness of our chemical constraints and topological consistency checks. These results establish MotifAgent as a new benchmark for AI-assisted reaction prediction, providing a novel technical pathway for computational chemistry applications.

Table 6: Performance comparison on reaction prediction tasks.

| Method | EXACT↑ | BLEU↑ | LEVENSHTEIN↓ | RDK FTS↑ | MACCS FTS↑ | MORGAN FTS↑ | VALIDITY↑ |
|---|---|---|---|---|---|---|---|
| *Reagent Prediction* | | | | | | | |
| Alpaca | 0.000 | 0.026 | 29.037 | 0.029 | 0.016 | 0.001 | 0.186 |
| Baize | 0.000 | 0.051 | 30.628 | 0.022 | 0.018 | 0.004 | 0.099 |
| ChatGLM | 0.000 | 0.019 | 29.169 | 0.017 | 0.006 | 0.002 | 0.074 |
| LLama | 0.000 | 0.003 | 28.040 | 0.037 | 0.001 | 0.001 | 0.001 |
| Vicuna | 0.000 | 0.010 | 27.948 | 0.038 | 0.002 | 0.001 | 0.007 |
| Mol-Instruction | 0.044 | 0.224 | 23.167 | 0.237 | 0.364 | 0.213 | 1.000 |
| Llama-7b (LoRA) | 0.000 | 0.283 | 53.510 | 0.136 | 0.294 | 0.106 | 1.000 |
| InstructMol-G | 0.031 | 0.429 | 31.447 | 0.389 | 0.249 | 0.220 | 1.000 |
| InstructMol-GS | 0.057 | 0.439 | 29.757 | 0.437 | 0.314 | 0.271 | 0.999 |
| HIGHT-G | 0.050 | 0.462 | 28.970 | 0.441 | 0.314 | 0.275 | 1.000 |
| HIGHT-GS | 0.067 | 0.482 | 27.167 | 0.462 | 0.346 | 0.303 | 1.000 |
| MotifAgent | **0.085** | **0.516** | **22.571** | **0.502** | **0.376** | **0.379** | **1.000** |
| *Forward Reaction Prediction* | | | | | | | |
| Alpaca | 0.000 | 0.065 | 41.989 | 0.004 | 0.024 | 0.008 | 0.138 |
| Baize | 0.000 | 0.044 | 41.500 | 0.004 | 0.025 | 0.009 | 0.097 |
| ChatGLM | 0.000 | 0.183 | 40.008 | 0.050 | 0.100 | 0.044 | 0.108 |
| LLama | 0.000 | 0.020 | 42.002 | 0.001 | 0.002 | 0.001 | 0.039 |
| Vicuna | 0.000 | 0.057 | 41.690 | 0.007 | 0.016 | 0.006 | 0.059 |
| Mol-Instruction | 0.045 | 0.654 | 27.262 | 0.313 | 0.509 | 0.262 | 1.000 |
| Llama-7b (LoRA) | 0.012 | 0.804 | 29.947 | 0.499 | 0.649 | 0.407 | 1.000 |
| InstructMol-G | 0.031 | 0.853 | 24.790 | 0.512 | 0.362 | 0.303 | 0.993 |
| InstructMol-GS | 0.252 | 0.926 | 17.773 | 0.755 | 0.599 | 0.543 | 1.000 |
| HIGHT-G | 0.037 | 0.869 | 23.759 | 0.590 | 0.394 | 0.340 | 0.993 |
| HIGHT-GS | 0.293 | 0.935 | 16.687 | 0.774 | 0.618 | 0.566 | 1.000 |
| MotifAgent | **0.315** | **0.937** | **15.127** | **0.806** | **0.669** | **0.582** | **1.000** |
| *Retrosynthesis* | | | | | | | |
| Alpaca | 0.000 | 0.063 | 46.915 | 0.005 | 0.023 | 0.007 | 0.160 |
| Baize | 0.000 | 0.095 | 44.714 | 0.025 | 0.050 | 0.023 | 0.112 |
| ChatGLM | 0.000 | 0.117 | 48.365 | 0.056 | 0.075 | 0.043 | 0.046 |
| LLama | 0.000 | 0.036 | 46.844 | 0.018 | 0.029 | 0.017 | 0.010 |
| Vicuna | 0.000 | 0.057 | 46.877 | 0.025 | 0.030 | 0.021 | 0.017 |
| Mol-Instruction | 0.009 | 0.705 | 31.227 | 0.283 | 0.487 | 0.230 | 1.000 |
| Llama-7b (LoRA) | 0.000 | 0.283 | 53.510 | 0.136 | 0.294 | 0.106 | 1.000 |
| InstructMol-G | 0.001 | 0.835 | 31.359 | 0.447 | 0.277 | 0.241 | 0.996 |
| InstructMol-GS | 0.172 | 0.911 | 20.300 | 0.765 | 0.615 | 0.568 | 1.000 |
| HIGHT-G | 0.008 | 0.863 | 28.912 | 0.564 | 0.340 | 0.309 | 1.000 |
| HIGHT-GS | 0.202 | 0.914 | 20.194 | 0.772 | 0.623 | 0.577 | 0.999 |
| MotifAgent | **0.275** | **0.932** | **18.810** | **0.783** | **0.685** | **0.631** | **1.000** |

# C  ALGORITHM

# D  EVALUATION METRICS

This section provides detailed descriptions of all evaluation metrics used in our ablation studies, including their definitions and computational procedures.

## D.1  MULTI-AGENT VS. SINGLE-AGENT ARCHITECTURE METRICS

**Reconstruction Accuracy (%)** measures the percentage of test molecules that are perfectly reconstructed with identical topology to the target. A molecule is considered successfully reconstructed when the assembled molecular graph has the same node set (all motifs included), identical edge set (all connections match), and forms a single connected component. We compute this as the ratio of perfectly reconstructed molecules to total test molecules, multiplied by 100.

**Chemical Validity (%)** evaluates the percentage of assembled molecules that satisfy fundamental chemical constraints. For each assembled molecule, we verify valence rules for all atoms, check for proper aromaticity preservation, ensure no sterically impossible connections exist, and validate that all formed rings are chemically reasonable. A molecule passes validity checks only when all constraints are satisfied. The metric is computed as the ratio of chemically valid assemblies to total assembly attempts.

---

**Algorithm 1** MotifAgent: Multi-Agent Molecular Assembly with Topological Learning

---

**Require:** Molecule dataset $\mathcal{D}$, Target properties $\mathcal{Y}$
**Ensure:** Trained policy $\pi_\theta$, Critic $V_\phi$
 1: **function** FRAGMENTMOLECULE($M$)
 2:     $\mathcal{M}, E^* \leftarrow$ BRICS($M$); $\mathcal{T} \leftarrow$ TextSerialize($\mathcal{M}$)
 3:     **return** $\mathcal{M}, \mathcal{T}, E^*$
 4: **end function**
 5: **function** MULTIAGENTPROPOSAL($\mathcal{A}, G_t$)
 6:     $\mathcal{P} \leftarrow \emptyset$
 7: **for** each agent $a_i \in \mathcal{A}$ **do**
 8:         $o_i \leftarrow$ LocalObs($a_i, G_t$); $h_{\text{topo}} \leftarrow$ GlobalTopo($G_t$)
 9:         $\mathcal{P} \leftarrow \mathcal{P} \cup \{\pi_\theta(o_i, h_{\text{topo}})\}$ {Hierarchical sampling}
10: **end for**
11:     **return** $\mathcal{P}$
12: **end function**
13: **function** CENTRALARBITRATION($\mathcal{P}, E^*, G_t$, stage)
14:     $\mathcal{P}_{\text{valid}} \leftarrow$ ChemFilter($\mathcal{P}$)
15: **if** stage $\leq 2$ **then**
16:         $\mathcal{P}_{\text{valid}} \leftarrow$ TargetMask($\mathcal{P}_{\text{valid}}, E^*$)
17: **end if**
18:     **return** $\arg\max_{a \in \mathcal{P}_{\text{valid}}} [w_1 \cdot \text{Chem}(a) + w_2 \cdot \text{Topo}(a) + w_3 \cdot \text{Prop}(a)]$
19: **end function**
20: **function** OPTIMIZE($\mathcal{B}, \pi_\theta, V_\phi, \pi_{\text{ref}}, E^*$)
21:     // Compute advantages and returns
22:     $\hat{A} \leftarrow \sum_{l=0}^{\infty} (\gamma\lambda)^l \delta_{t+l}$; $\hat{R} \leftarrow$ MonteCarloReturn($\mathcal{B}$)
23:     // Actor loss with Set-BC
24:     $\mathcal{A}^* \leftarrow \{a: \text{preserves target edges from } E^*\}$
25:     $\mathcal{L}_{\text{actor}} \leftarrow -\mathcal{L}_{\text{PPO}} - \beta\mathcal{H}(\pi_\theta) + \alpha_{BC} \log \sum_{a \in \mathcal{A}^*} \pi_\theta(a|s) + \beta_{KL}\text{KL}(\pi_\theta || \pi_{\text{ref}})$
26:     // Critic loss with auxiliary tasks
27:     $\mathcal{L}_{\text{critic}} \leftarrow (V_\phi - \hat{R})^2 + w_1(V_{\text{edges}} - |E^* \setminus E_t|)^2 + w_2(V_{\text{cc}} - \text{cc}(G_t))^2$
28:     Update $\theta \leftarrow \theta - \eta_\pi \nabla_\theta \mathcal{L}_{\text{actor}}$; Update $\phi \leftarrow \phi - \eta_V \nabla_\phi \mathcal{L}_{\text{critic}}$
29: **end function**
30: // **Main Training Loop**
31: $\pi_\theta \leftarrow$ SharedLLM(); $V_\phi \leftarrow$ Critic(); $\pi_{\text{ref}} \leftarrow \pi_\theta$; stage $\leftarrow 1$
32: **for** episode $= 1$ to $N$ **do**
33:     $M \sim \mathcal{D}$; $\mathcal{M}, \mathcal{T}, E^* \leftarrow$ FRAGMENTMOLECULE($M$)
34:     $\mathcal{A} \leftarrow$ [MotifAgent($m, \pi_\theta$) for $m \in \mathcal{M}$]; $G_t \leftarrow \emptyset$
35:     **while** $\neg$Terminal($G_t, E^*$) **do**
36:         $\mathcal{P} \leftarrow$ MULTIAGENTPROPOSAL($\mathcal{A}, G_t$)
37:         $a_t \leftarrow$ CENTRALARBITRATION($\mathcal{P}, E^*, G_t$, stage)
38:         $G_{t+1}, r_t \leftarrow$ Execute($G_t, a_t$); $r_t \leftarrow r_t + \gamma\Phi(G_{t+1}) - \Phi(G_t)$
39:         $\mathcal{B} \leftarrow \mathcal{B} \cup \{(G_t, a_t, r_t, G_{t+1})\}$; $G_t \leftarrow G_{t+1}$
40:     **end while**
41:     **if** $|\mathcal{B}| \geq$ batch_size **then**
42:         OPTIMIZE($\mathcal{B}, \pi_\theta, V_\phi, \pi_{\text{ref}}, E^*$); $\mathcal{B} \leftarrow \emptyset$
43:         **if** performance meets criteria **then** stage $\leftarrow$ stage $+1$; Adjust($\alpha_{BC}, w_i$)
44:     **end if**
45: **end for**

---

**Graph Edit Distance** quantifies the structural difference between the assembled molecule and the target molecule using the minimum number of graph edit operations (node insertion, node deletion, edge insertion, edge deletion) required to transform one molecular graph into another. Lower values indicate better topological reconstruction. We use the Hungarian algorithm for optimal node matching and compute the edit distance considering both node labels (motif types) and edge labels (bond types).

**Morgan Fingerprint Tanimoto Similarity (Morgan FTS)** measures the structural similarity between assembled and target molecules using Morgan fingerprints with radius 2 and 2048 bits. The Tanimoto coefficient is calculated as the ratio of the intersection to the union of fingerprint bits. Values range from 0 (completely dissimilar) to 1 (identical), with higher values indicating better preservation of local structural features and chemical environments.

**Connection Site Accuracy (%)** evaluates the model's ability to identify correct connection points on motifs. For each successful connection made during assembly, we check whether the chosen sites on both motifs match the sites used in the target molecule. The metric is computed as the percentage of connections where both source and target sites are correctly identified, regardless of bond type.

**Bond Type Accuracy (%)** measures the correctness of bond type selection given that the connection sites are correct. Among all connections with correctly identified sites, we calculate the percentage where the chosen bond type (single, double, triple, aromatic) matches the target. This metric isolates the model's understanding of chemical bonding rules from its ability to identify connection sites.

## D.2 SET-BC VS. FIXED-ORDER SUPERVISION METRICS

**Average Assembly Steps** measures the mean number of connection actions required to successfully complete molecular assembly, excluding failed attempts. For each successfully assembled molecule, we count the number of connection actions (not including STOP actions) from the initial state to completion. Lower values indicate more efficient assembly strategies, as the model discovers shorter paths to construct the target molecule.

**Path Entropy** quantifies the diversity of assembly strategies learned by the model. For each target molecule, we generate 100 independent assembly trajectories and encode each as a sequence of connection actions. We then compute the Shannon entropy over the distribution of unique assembly paths: $H = -\sum_i p_i \log p_i$, where $p_i$ is the frequency of the $i$-th unique path. Higher entropy indicates the model has learned multiple valid assembly strategies, while lower entropy suggests convergence to a limited set of paths or a single dominant strategy.

**Sample Efficiency - Performance with Limited Data** evaluates model robustness under data-scarce conditions. We train models using randomly sampled subsets of the training data (10%, 25%, 50%, and 100%) while keeping the test set fixed. For each data regime, we report the Overall Success Rate, which combines reconstruction accuracy and chemical validity: a molecule is considered successful only if it is both perfectly reconstructed and chemically valid. This metric reveals how effectively each approach leverages limited training examples, with larger performance gaps in low-data regimes indicating superior sample efficiency.

The Path Diversity Analysis protocol involves generating multiple trajectories for identical targets to assess whether the model has learned the flexibility inherent in molecular assembly. Each of the 100 independent runs starts from the same initial state (disconnected motifs) but uses different random seeds for action sampling. The resulting path distribution reveals whether the model treats assembly as a rigid sequence (low diversity) or understands the equivalence of multiple assembly orders (high diversity).

## E FUTURE WORK

Future work should focus on developing more efficient model architectures to reduce computational requirements while maintaining performance. This includes exploring lightweight multi-agent designs, knowledge distillation techniques, and more efficient training strategies. Additionally, expanding the approach to handle larger molecular systems through hierarchical decomposition or progressive assembly strategies would significantly broaden its applicability. Validation on larger, more diverse datasets covering broader chemical space, including natural products, polymers, and

organometallic compounds, would provide stronger evidence of the method's generalizability and robustness across different molecular classes.

# F    CASES STUDY

In this section, we demonstrate the conversational and decision-making processes of MotifAgent through four representative cases. We first present a standard Algorithm Demonstration using the aspirin molecule, followed by detailed assembly workflows on three molecules of varying complexity: a simple molecule (Paracetamol), a complex molecule (Ibuprofen), and a complex heterocyclic molecule (Omeprazole).

Given the input molecular SMILES, target properties, and fragmentation results, each motif establishes its individual profile and analyzes its current state and relationships with other motifs. Through chemical validity assessment and topological scoring, agents evaluate the benefits of establishing inter-motif connections, execute connection actions, and compute corresponding rewards. After multiple rounds of negotiation, all motifs are successfully connected, followed by a comprehensive validation to ensure molecular validity and property satisfaction. The process concludes with the output of the final assembled molecule.

## Case 1: Algorithm Demonstration: Multi-Agent Assembly of Aspirin

**Input Molecule**

**Initial Setup**
**Target Molecule :** Aspirin ((Acetylsalicylic acid))
**SMILES :** CC(=O)Oc1ccccc1C(=O)O
**Target Properties:** Anti-inflammatory (COX inhibition), LogP ≈ 1.2, MW = 180.16
**Fragmentation Results:**
**Motif 1:** Benzene ring with two substituent sites (c1ccccc1, 6 atoms)
**Motif 2:** Carboxyl group (C(=O)O, 3 atoms)
**Motif 3:** Acetyl group (CC(=O), 3 atoms)
**Motif 4:** Oxygen linker (O, 1 atom, acts as bridge)

**Episode Start: Multi-Agent Dialogue**

**Environment State (t=0):**
Current Assembly: Empty graph
Connected Components: 0
Available Motifs: [Motif_1, Motif_2, Motif_3, Motif_4]
Target Edges: 3 (benzene-carboxyl, benzene-oxygen, oxygen-acetyl)
Current Properties: None

**Motif 1**  **Motif 2**
**Motif 3**  **Motif 4**

Global Coordinator

Agents, we need to assemble **aspirin**. The target has **anti-inflammatory** properties with **LogP around 1.2**. **Current topology is empty.** Please propose your initial connections.

**Round 1: Initial Core Selection**

**Motif 1 (Benzene)**
**My Structure:** Aromatic ring, 6 carbons
**Available Sites:** [ortho_1, ortho_2, meta_1, meta_2, para]
**Chemical Context:** I'm a stable aromatic core, ideal as scaffold
**Proposal:** I should be the starting point. Ortho substitution pattern typically yields aspirin-like properties.
**Priority:** HIGH - I'm the largest motif and central scaffold

**Motif 2 (Carboxyl)**
**My Structure:** C(=O)O, acidic functional group
**Available Sites:** [carbon_connection]
**Chemical Context:** I provide acidity (pKa ~3-4) crucial for COX binding
**Proposal:** Connect to Agent_1's ortho position for salicylic acid core
**Reasoning:** Ortho-hydroxybenzoic acid is the pharmacophore base
**Priority:** HIGH - Essential for biological activity

**Motif 3 (Acetyl)**
**My Structure:** CC(=O), ester-forming group
**Available Sites:** [carbonyl_carbon]
**Chemical Context:** I can form esters, improving stability over free phenol
**Proposal:** Wait for oxygen bridge, then form ester linkage
**Reasoning:** Direct connection to benzene would create ketone, not ester
**Priority:** MEDIUM - Need linker first

**Motif 4 (Oxygen)**
**My Structure:** Single oxygen atom, bridging unit
**Available Sites:** [connection_1, connection_2]
**Chemical Context:** I enable ester formation, crucial for aspirin vs salicylic acid
**Proposal:** Bridge between Agent_1 and Agent_3 after carboxyl attachment
**Reasoning:** Ester linkage masks the phenol, reducing GI irritation
**Priority:** MEDIUM - Sequential dependency

**Central Arbitrator Evaluation**
**Chemical Validity Check:**
- Benzene-Carboxyl (ortho): ✓ Valid, maintains aromaticity
- Benzene-Oxygen (direct): ✓ Valid, forms phenolic ether
- Acetyl-Oxygen: ✓ Valid, forms ester
**Topological Scoring:**
S(benzene→carboxyl) = 0.8 (stability) + 0.9 (progress) + 0.7 (property) = 2.4
S(benzene→oxygen) = 0.7 (stability) + 0.6 (progress) + 0.3 (property) = 1.6
**Decision: Execute benzene-carboxyl connection first**

**Reward Calculation:**
r_valid = +1.0 (valid connection)
r_stable = -0.1 (minimal strain)
r_func = +0.5 (formed benzoic acid substructure)
r_conn = +1.0 (created first connected component)
r_edge = +1.0 (matched target edge)
**Total: R = 3.4**

**Action Executed:** Connect: Motif_1[ortho_1] --single--> Motif_2[carbon]

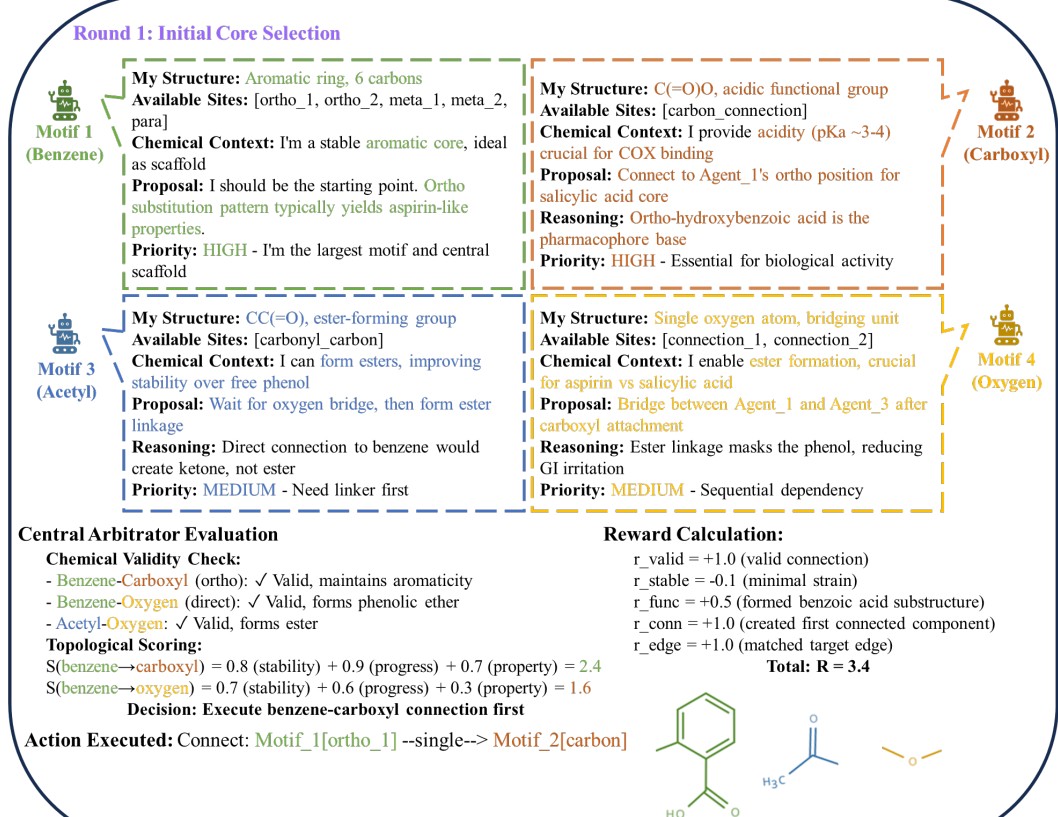

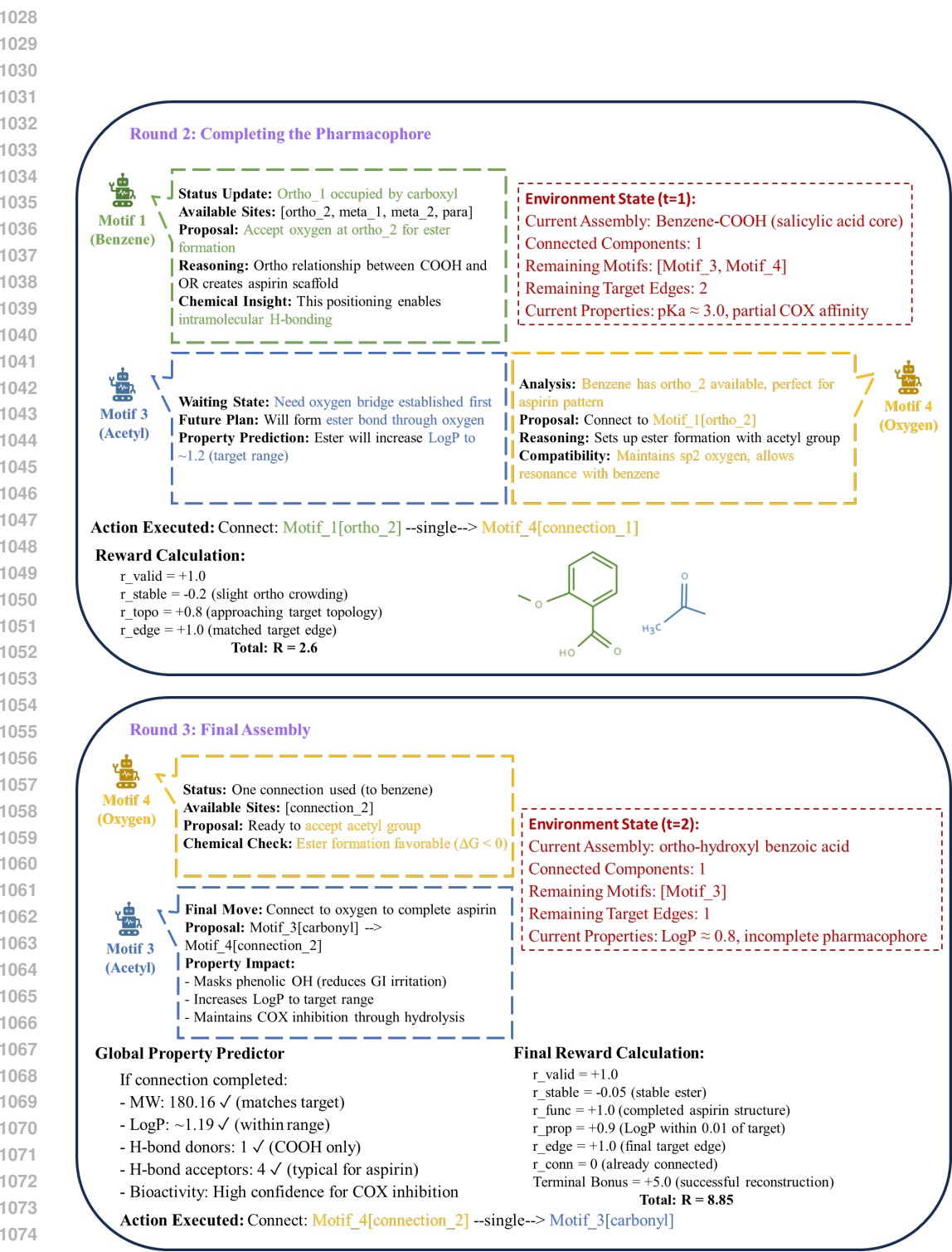

**Round 2: Completing the Pharmacophore**

**Motif 1 (Benzene)**

**Status Update:** Ortho_1 occupied by carboxyl
**Available Sites:** [ortho_2, meta_1, meta_2, para]
**Proposal:** Accept oxygen at ortho_2 for ester formation
**Reasoning:** Ortho relationship between COOH and OR creates aspirin scaffold
**Chemical Insight:** This positioning enables intramolecular H-bonding

**Environment State (t=1):**
Current Assembly: Benzene-COOH (salicylic acid core)
Connected Components: 1
Remaining Motifs: [Motif_3, Motif_4]
Remaining Target Edges: 2
Current Properties: pKa ≈ 3.0, partial COX affinity

**Motif 3 (Acetyl)**

**Waiting State:** Need oxygen bridge established first
**Future Plan:** Will form ester bond through oxygen
**Property Prediction:** Ester will increase LogP to ~1.2 (target range)

**Analysis:** Benzene has ortho_2 available, perfect for aspirin pattern
**Proposal:** Connect to Motif_1[ortho_2]
**Reasoning:** Sets up ester formation with acetyl group
**Compatibility:** Maintains sp2 oxygen, allows resonance with benzene

**Motif 4 (Oxygen)**

**Action Executed:** Connect: Motif_1[ortho_2] --single--> Motif_4[connection_1]

**Reward Calculation:**
r_valid = +1.0
r_stable = -0.2 (slight ortho crowding)
r_topo = +0.8 (approaching target topology)
r_edge = +1.0 (matched target edge)
**Total: R = 2.6**

**Round 3: Final Assembly**

**Motif 4 (Oxygen)**

**Status:** One connection used (to benzene)
**Available Sites:** [connection_2]
**Proposal:** Ready to accept acetyl group
**Chemical Check:** Ester formation favorable (ΔG < 0)

**Environment State (t=2):**
Current Assembly: ortho-hydroxyl benzoic acid
Connected Components: 1
Remaining Motifs: [Motif_3]
Remaining Target Edges: 1
Current Properties: LogP ≈ 0.8, incomplete pharmacophore

**Motif 3 (Acetyl)**

**Final Move:** Connect to oxygen to complete aspirin
**Proposal:** Motif_3[carbonyl] --> Motif_4[connection_2]
**Property Impact:**
- Masks phenolic OH (reduces GI irritation)
- Increases LogP to target range
- Maintains COX inhibition through hydrolysis

**Global Property Predictor**

If connection completed:
- MW: 180.16 ✓ (matches target)
- LogP: ~1.19 ✓ (within range)
- H-bond donors: 1 ✓ (COOH only)
- H-bond acceptors: 4 ✓ (typical for aspirin)
- Bioactivity: High confidence for COX inhibition

**Final Reward Calculation:**
r_valid = +1.0
r_stable = -0.05 (stable ester)
r_func = +1.0 (completed aspirin structure)
r_prop = +0.9 (LogP within 0.01 of target)
r_edge = +1.0 (final target edge)
r_conn = 0 (already connected)
Terminal Bonus = +5.0 (successful reconstruction)
**Total: R = 8.85**

**Action Executed:** Connect: Motif_4[connection_2] --single--> Motif_3[carbonyl]

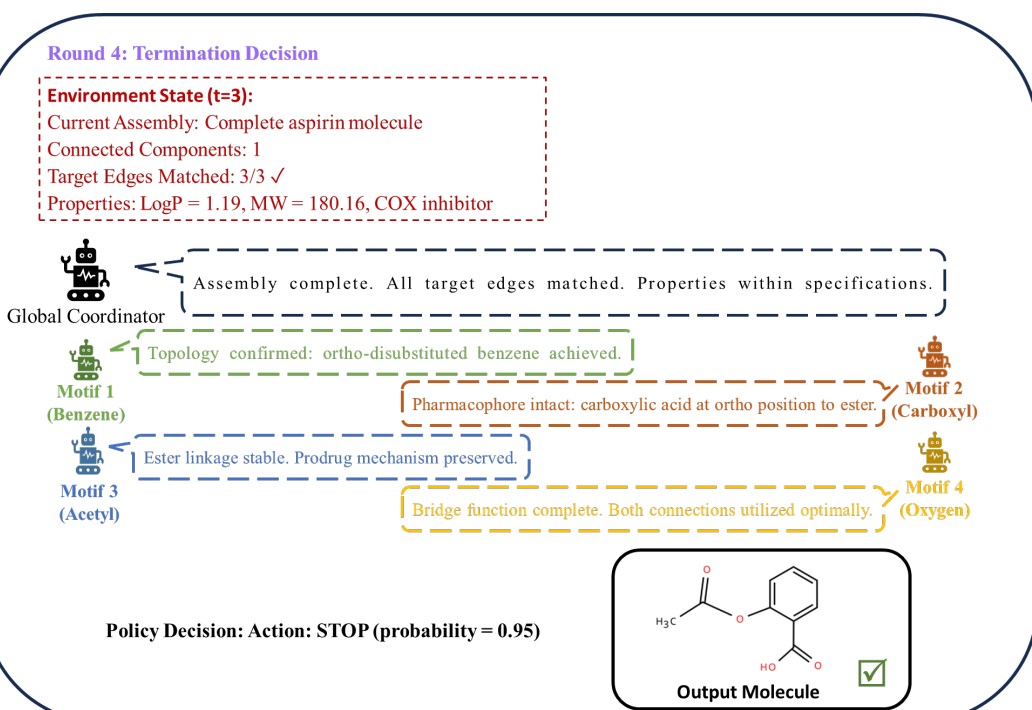

**Round 4: Termination Decision**

**Environment State (t=3):**
Current Assembly: Complete aspirin molecule
Connected Components: 1
Target Edges Matched: 3/3 ✓
Properties: LogP = 1.19, MW = 180.16, COX inhibitor

Global Coordinator

Assembly complete. All target edges matched. Properties within specifications.

Topology confirmed: ortho-disubstituted benzene achieved.

**Motif 1
(Benzene)**

**Motif 2
(Carboxyl)**

Pharmacophore intact: carboxylic acid at ortho position to ester.

Ester linkage stable. Prodrug mechanism preserved.

**Motif 3
(Acetyl)**

**Motif 4
(Oxygen)**

Bridge function complete. Both connections utilized optimally.

**Policy Decision: Action: STOP (probability = 0.95)**

**Output Molecule**

**Episode Summary**

**Trajectory Interpretation:**
Step 1: Established salicylic acid core (benzene-COOH)
      → Created primary pharmacophore
Step 2: Added oxygen at ortho position
      → Prepared ester formation site
Step 3: Completed acetyl ester
      → Achieved target molecule with desired properties

**Key Insights Learned:**
- Ortho substitution pattern critical for aspirin activity
- Assembly order: scaffold → functional groups → modifiers
- Ester formation requires bridging oxygen (not direct connection)
- Properties emerge from specific topology (ortho relationship)

**Chemical Explanation Generated**: The assembly successfully reconstructed aspirin through strategic ortho-substitution on benzene. The carboxyl group provides COX binding affinity while the acetyl ester masks the phenolic hydroxyl, reducing gastric irritation while maintaining efficacy through in vivo hydrolysis. The learned assembly pattern prioritizes pharmacophore establishment before property optimization.

## Case 2: Paracetamol (Acetaminophen) - Simple Molecule

**Input Molecule**

**Initial Setup**
**Target Molecule :** Paracetamol
**SMILES :** CC(=O)Nc1ccc(O)cc1
**Target Properties:** Analgesic/antipyretic, LogP ≈ 0.46, MW = 151.16
**PubChem CID:** 1983
**Fragmentation Results**:

**Motif 1**: para-disubstituted benzene (c1ccc(-)cc1, 6 atoms)
**Motif 2**: Hydroxyl group (O, 1 atom)
**Motif 3**: Acetamide group (CC(=O)N, 4 atoms)

**Episode Start: Multi-Agent Dialogue**

**Environment State (t=0):**
Current Assembly: Empty graph
Connected Components: 0
Available Motifs: [Motif_1, Motif_2, Motif_3]
Target Edges: 2 (benzene-hydroxyl, benzene-acetamide)
Current Properties: None
Target: Analgesic with low LogP for CNS penetration

**Motif 1**   **Motif 2**

**Motif 3**

Global Coordinator

Agents, we need to assemble **paracetamol**. The target has **analgesic properties** with **LogP around 0.46**. Current topology is empty. Please propose your initial connections.

**Round 1: Initial Core Selection**

**Motif 1 (Benzene)**
**My Structure:** Aromatic ring, 6 carbons, para-substitutable
**Available Sites:** [para_1, para_2, ortho_1, ortho_2, meta_1, meta_2]
**Chemical Context:** I'm the scaffold for most analgesics
**Proposal:** I should be the starting point. Para substitution reduces toxicity compared to ortho/meta patterns
**Priority:** HIGH - I'm the core scaffold

**Motif 2 (Hydroxyl)**
**My Structure:** Single OH group
**Available Sites:** [oxygen_connection]
**Chemical Context:** I provide H-bonding for receptor interaction
**Proposal:** Connect to Agent_1's para position for phenolic structure
**Reasoning:** Phenol group essential for COX interaction
**Priority:** HIGH - Key pharmacophore element

**Motif 3 (Acetamide)**
**My Structure:** CC(=O)N, metabolically stable group
**Available Sites:** [nitrogen_connection]
**Chemical Context**: I provide metabolic stability vs acetyl ester
**Proposal:** Connect to Agent_1's other para position
**Reasoning:** Amide more stable than ester in vivo
**Priority:** HIGH - Prevents toxic metabolite formation

**Central Arbitrator Evaluation**
**Chemical Validity Check:**
- Benzene-Hydroxyl (para): ✓ Valid, forms p-hydroxyphenyl
- Benzene-Acetamide (para): ✓ Valid, forms p-acetamidophenyl
**Topological Scoring:**
S(benzene→hydroxyl) = 0.9 (stability) + 0.8 (progress) + 0.9 (property) = 2.6
S(benzene→acetamide) = 0.8 (stability) + 0.8 (progress) + 0.7 (property) = 2.3
**Decision: Execute benzene-hydroxyl connection first**

**Action Executed:** Connect: Motif_1[para_1] --single--> Motif_2[oxygen]

**Reward Calculation:**
r_valid = +1.0 (valid connection)
r_stable = -0.05 (minimal strain, para position)
r_func = +0.8 (formed phenolic group)
r_conn = +1.0 (created first connected component)
r_edge = +1.0 (matched target edge)
**Total: R = 3.75**

**Round 2: Completing the Structure**

**Environment State (t=1):**
Current Assembly: p-hydroxyphenyl
Connected Components: 1
Remaining Motifs: [Motif_3]
Remaining Target Edges: 1
Current Properties: pKa ≈ 10 (phenolic), partial analgesic activity

**Motif 1 (Benzene)**

**Status Update:** para_1 occupied by hydroxyl
**Available Sites:** [para_2, ortho_1, ortho_2, meta_1, meta_2]
**Proposal:** Accept acetamide at para_2
**Reasoning:** Creates p-hydroxyacetanilide (paracetamol)
**Chemical Insight:** Electron donation from both groups activates ring safely

**Analysis:** Para position to hydroxyl is optimal
**Proposal:** Connect to Motif_1[para_2]
**Property Prediction:** Will achieve target LogP ~0.46
**Mechanism:** Amide prevents formation of toxic NAPQI at normal doses

**Motif 3 (Acetamide)**

**Action Executed:** Connect: Motif_1[para_2] --single--> Motif_3[nitrogen]

**Reward Calculation:**
r_valid = +1.0
r_stable = -0.05 (no steric clash in para)
r_func = +1.0 (completed paracetamol structure)
r_prop = +0.9 (LogP = 0.47, within target)
r_edge = +1.0 (matched final target edge)
Terminal Bonus = +5.0 (successful reconstruction)
**Total: R = 8.85**

**Round 3: Termination Decision**

**Environment State (t=2):**
Current Assembly: Complete paracetamol molecule
Connected Components: 1
Target Edges Matched: 2/2 ✓
Properties: LogP = 0.47, MW = 151.16, pKa = 9.38

Global Coordinator

Assembly complete. All target edges matched. Properties within specifications.

**Policy Decision: Action: STOP (probability = 0.96)**

**Chemical Explanation Generated**: Paracetamol assembled through para-substitution pattern. The phenolic hydroxyl and acetamide groups in para positions provide optimal balance of activity and safety, avoiding toxic quinone imine formation seen with other substitution patterns.

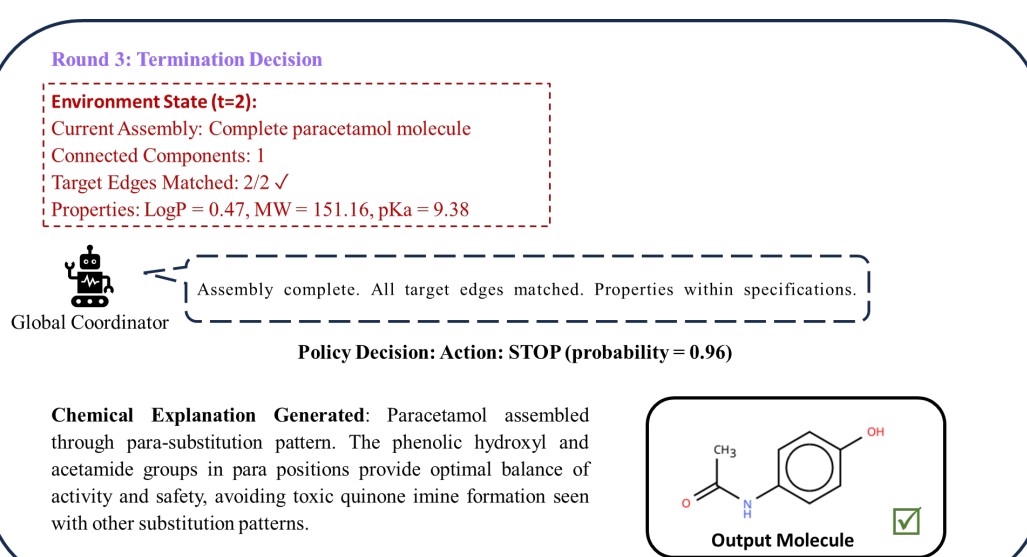

**Output Molecule**

## Case 3: Ibuprofen - Complex Molecule

**Initial Setup**
**Target Molecule :** Ibuprofen
**SMILES :** CC(C)Cc1ccc(C(C)C(=O)O)cc1
**Target Properties:** NSAID, LogP ≈ 3.97, MW = 206.28
**PubChem CID:** 3672
**Fragmentation Results:**
**Motif 1**: para-disubstituted benzene (c1ccc(-)cc1, 6 atoms)
**Motif 2**: Isobutyl group (CC(C)C, 4 atoms)
**Motif 3**: α-methylpropionic acid (C(C)C(=O)O, 4 atoms)

**Episode Start: Multi-Agent Dialogue**

**Environment State (t=0):**
Current Assembly: Empty graph
Connected Components: 0
Available Motifs: [Motif_1, Motif_2, Motif_3]
Target Edges: 2 (benzene-isobutyl, benzene-propionic)
Current Properties: None
Target: NSAID with high LogP for membrane penetration

**Motif 1**   **Motif 2**

**Motif 3**

Global Coordinator

Agents, we need to assemble **ibuprofen**. The target is an **NSAID** with **LogP around 3.97** for **optimal membrane penetration**. Please propose your initial connections.

**Round 1: Initial Core Selection**

**Motif 1 (Benzene)**
**My Structure:** Aromatic ring with para substitution sites
**Available Sites:** [para_1, para_2, ortho_1, ortho_2, meta_1, meta_2]
**Chemical Context:** Central scaffold for propionic acid NSAIDs
**Proposal:** I should anchor the assembly
**Priority:** HIGH - I'm the core connecting both alkyl groups

**Motif 2 (Isobutyl)**
**My Structure:** CC(C)C, branched alkyl chain
**Available Sites:** [primary_carbon]
**Chemical Context:** I increase lipophilicity and membrane permeability
**Proposal:** Connect to benzene's para position
**Reasoning:** Creates hydrophobic pocket for COX active site
**Priority:** MEDIUM - Important for selectivity but not primary pharmacophore

**Motif 3 (α-Methylpropionic acid)**
**My Structure:** C(C)C(=O)O, chiral center at α-position
**Available Sites:** [alpha_carbon]
**Chemical Context:** I'm the primary pharmacophore for COX inhibition
**Proposal:** Connect to benzene para (opposite to isobutyl)
**Reasoning:** α-methyl provides COX-2 selectivity
**Priority:** HIGH - Essential for NSAID activity

**Central Arbitrator Evaluation**
**Chemical Validity Check:**
- Benzene-Isobutyl: ✓ Valid, increases lipophilicity
- Benzene-Propionic: ✓ Valid, forms arylpropionic acid
**Topological Scoring:**
S(benzene→propionic) = 0.8 (stability) + 0.9 (progress) + 1.0 (property) = 2.7
S(benzene→isobutyl) = 0.9 (stability) + 0.7 (progress) + 0.6 (property) = 2.2
**Decision: Execute benzene-propionic connection first**

**Reward Calculation:**
r_valid = +1.0 (valid connection)
r_stable = -0.1 (slight steric from α-methyl)
r_func = +1.0 (formed arylpropionic acid pharmacophore)
r_conn = +1.0 (created first connected component)
r_edge = +1.0 (matched target edge)
**Total: R = 3.9**

**Action Executed:** Connect: Motif_1[para_1] --single--> Motif_3[alpha_carbon]

**Round 2: Completing Lipophilic Modification**

**Environment State (t=1):**
Current Assembly: p-methylphenylpropionic acid
Connected Components: 1
Remaining Motifs: [Motif_2]
Remaining Target Edges: 1
Current Properties: pKa ≈ 4.5, LogP ≈ 2.1, partial COX activity

**Motif 1 (Benzene)**

**Status Update:** para_1 occupied by propionic acid
**Available Sites:** [para_2, ortho_1, ortho_2, meta_1, meta_2]
**Proposal:** Accept isobutyl at para_2
**Reasoning:** Para relationship minimizes steric clash
**Chemical Insight:** This completes the "profen" scaffold

**Analysis:** Para to propionic acid is ideal positioning
**Proposal:** Connect to Motif_1[para_2]
**Property Impact:** Will increase LogP to ~3.97 (target)
**COX Interaction:** Forms hydrophobic channel interaction

**Motif 2 (Isobutyl)**

**Global Property Predictor**

If connection completed:
- MW: 206.28 ✓ (matches target)
- LogP: ~3.96 ✓ (optimal for oral absorption)
- pKa: 4.4 ✓ (good tissue penetration)
- Rotatable bonds: 4 ✓ (acceptable flexibility)

**Reward Calculation:**
r_valid = +1.0
r_stable = -0.05 (minimal strain in para)
r_func = +0.5 (enhanced lipophilicity)
r_prop = +0.95 (LogP = 3.96, nearly perfect)
r_edge = +1.0 (matched final target edge)
Terminal Bonus = +5.0 (successful reconstruction)
**Total: R = 8.4**

**Action Executed:** Connect: Motif_1[para_2] --single--> Motif_2[primary_carbon]

**Round 3: Termination Decision**

**Environment State (t=2):**
Current Assembly: Complete ibuprofen molecule
Connected Components: 1
Target Edges Matched: 2/2 ✓
Properties: LogP = 3.96, MW = 206.28, COX-2 selective

Assembly complete. Ibuprofen structure achieved with optimal lipophilicity.

**Global Coordinator**

The α-methyl substitution pattern is crucial for reduced GI side effects compared to non-selective NSAIDs.

**Motif 3 (Propionic acid)**

**Policy Decision: Action: STOP (probability = 0.94)**

**Chemical Explanation Generated**: Ibuprofen assembled with para-disubstituted benzene pattern. The α-methylpropionic acid provides COX inhibition while the isobutyl group enhances membrane penetration and selectivity. The high LogP enables effective tissue distribution.

**Output Molecule**

## Case 4: Omeprazole - Complex Heterocyclic Molecule

**Input Molecule**

**Initial Setup**
**Target Molecule :** Omeprazole
**SMILES :** Oc1ccc2nc(S(=O)Cc3ncc(C)c(OC)c3C)[nH]c2c1
**Target Properties:** Proton pump inhibitor, LogP ≈ 2.23, MW = 345.42
**PubChem CID:** 4594
**Fragmentation Results:**
**Motif 1**: Benzimidazole core (c1ccc2[nH]c(-)nc2c1, 9 atoms)
**Motif 2**: Methoxy group (OC, 2 atoms)
**Motif 3**: 3-methylpyridine (c1ncc(C)cc1, 7 atoms)
**Motif 4**: Methoxy-methyl fragment (COC, 3 atoms)
**Motif 5**: Sulfinylmethyl bridge (S(=O)C, 3 atoms)

**Episode Start: Multi-Agent Dialogue**

**Environment State (t=0):**
Current Assembly: Empty graph
Connected Components: 0
Available Motifs: [Motif_1, Motif_2, Motif_3, Motif_4, Motif_5]
Target Edges: 4 (benzimidazole-methoxy, benzimidazole-sulfoxide,
        sulfoxide-pyridine, pyridine-methoxymethyl)
Current Properties: None
Target: PPI requiring acid activation, LogP ≈ 2.23

Motif 1    Motif 2    Motif 3    Motif 4    Motif 5

Global Coordinator

Agents, we need to assemble omeprazole with 5 motifs. This proton pump inhibitor requires precise heterocycle arrangement with a sulfoxide bridge. Please propose your initial connections.

**Round 1: Initial Core Selection**

**Motif 1 (Benzimidazole)**
**My Structure:** Fused heterocycle, 9 atoms
**Available Sites:** [C2_imidazole, C5_position]
**Chemical Context:** I'm the primary PPI pharmacophore, pKa ~6
**Proposal:** Start with me as foundation - need C5 methoxy for activity
**Reasoning:** My C2 position will later accept sulfoxide bridge
**Priority:** HIGH - Core pharmacophore

**Motif 2 (Methoxy)**
**My Structure:** OC, electron-donating group
**Available Sites:** [oxygen]
**Chemical Context:** I modulate benzimidazole electronics
**Proposal:** Attach to Motif_1 at C5 position first
**Reasoning:** Increases electron density for stability
**Priority:** HIGH - Essential for benzimidazole activation

**Motif 3 (3-Methylpyridine)**
**My Structure:** Pyridine with methyl at position 3
**Available Sites:** [C2, C4, C5]
**Chemical Context:** I need further decoration for full activity
**Proposal:** Wait - need my additional substituents
**Priority:** LOW - Require complete substitution pattern

**Motif 4 (Methoxy-methyl)**
**My Structure:** COC fragment
**Available Sites:** [carbon, oxygen]
**Chemical Context:** I complete pyridine substitution
**Proposal:** Will attach to pyridine later
**Priority:** MEDIUM - Needed for complete structure

**Motif 5 Sulfinyl methyl Bridge**
**My Structure:** S(=O)C, chiral sulfoxide with methyl
**Available Sites:** [sulfur, carbon]
**Chemical Context:** I'm the critical bridge and activation site
**Proposal:** Connect to benzimidazole after it's activated
**Reasoning:** Sulfoxide enables acid-triggered mechanism
**Priority:** HIGH - Essential bridge element

**Central Arbitrator Evaluation**

**Chemical Validity Check:**
- Benzimidazole-Methoxy (C5): ✓ Valid, enhances basicity
- Benzimidazole-Sulfoxide (C2): ✓ Valid, typical PPI pattern
**Topological Scoring:**
S(benzimidazole→methoxy) = 0.9 (stability) + 0.8 (progress) + 0.9 (property) = 2.6
S(benzimidazole alone) = 0.8 (stability) + 0.7 (progress) + 0.6 (property) = 2.1
        **Decision: Connect benzimidazole with methoxy first**

**Reward Calculation:**
$r_{valid}$ = +1.0 (valid connection)
$r_{stable}$ = -0.05 (minimal strain)
$r_{func}$ = +0.8 (enhanced benzimidazole basicity)
$r_{conn}$ = +1.0 (created first connected component)
$r_{edge}$ = +1.0 (matched target edge)
        **Total: R = 3.75**

**Action Executed:** Connect: Motif_1[C5_position] --single--> Motif_2[oxygen]

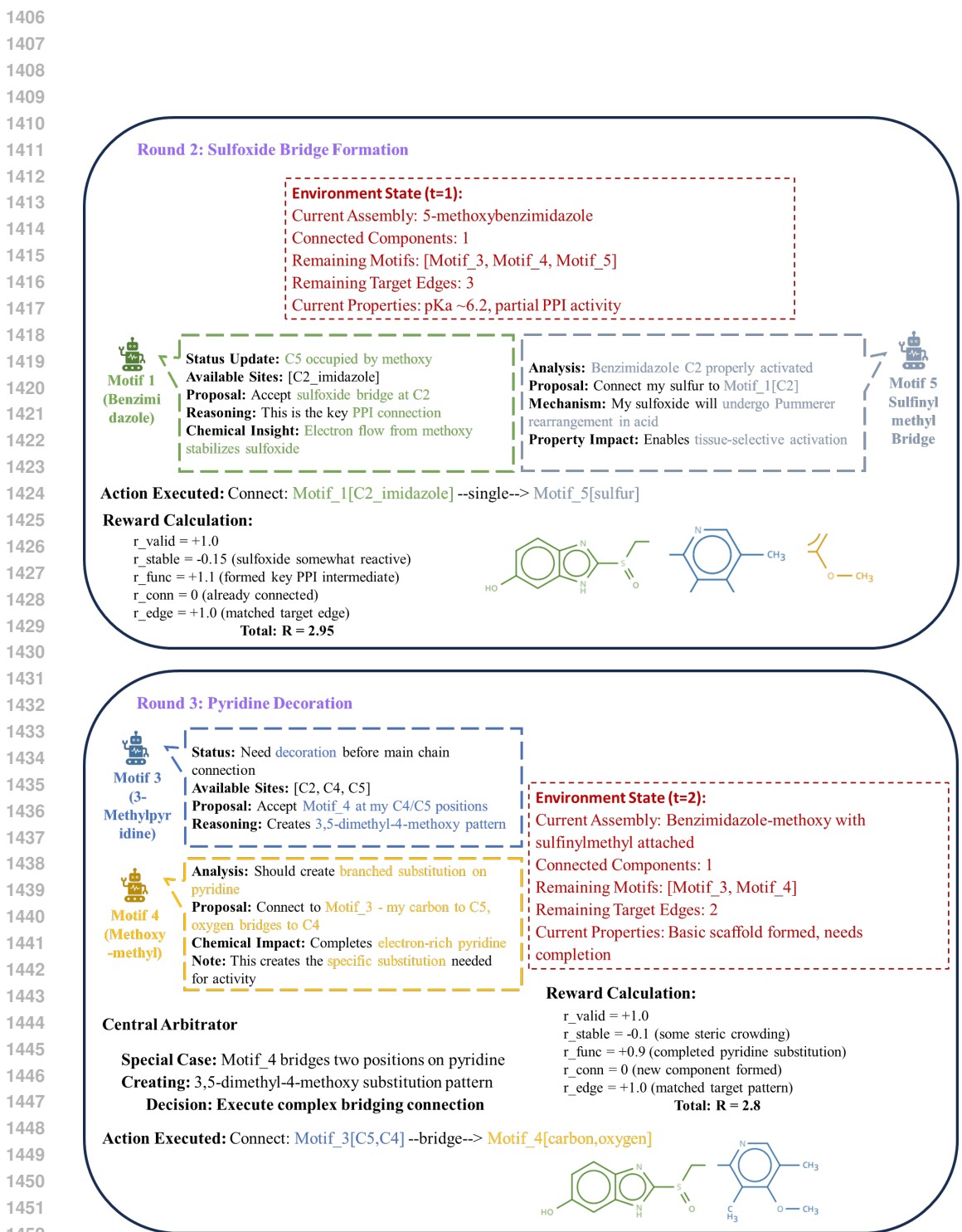

**Round 2: Sulfoxide Bridge Formation**

**Environment State (t=1):**
Current Assembly: 5-methoxybenzimidazole
Connected Components: 1
Remaining Motifs: [Motif_3, Motif_4, Motif_5]
Remaining Target Edges: 3
Current Properties: pKa ~6.2, partial PPI activity

**Motif 1 (Benzimi dazole)**

**Status Update:** C5 occupied by methoxy
**Available Sites:** [C2_imidazole]
**Proposal:** Accept sulfoxide bridge at C2
**Reasoning:** This is the key PPI connection
**Chemical Insight:** Electron flow from methoxy stabilizes sulfoxide

**Analysis:** Benzimidazole C2 properly activated
**Proposal:** Connect my sulfur to Motif_1[C2]
**Mechanism:** My sulfoxide will undergo Pummerer rearrangement in acid
**Property Impact:** Enables tissue-selective activation

**Motif 5 Sulfinyl methyl Bridge**

**Action Executed:** Connect: Motif_1[C2_imidazole] --single--> Motif_5[sulfur]

**Reward Calculation:**
r_valid = +1.0
r_stable = -0.15 (sulfoxide somewhat reactive)
r_func = +1.1 (formed key PPI intermediate)
r_conn = 0 (already connected)
r_edge = +1.0 (matched target edge)
**Total: R = 2.95**

**Round 3: Pyridine Decoration**

**Motif 3 (3-Methylpyr idine)**

**Status:** Need decoration before main chain connection
**Available Sites:** [C2, C4, C5]
**Proposal:** Accept Motif_4 at my C4/C5 positions
**Reasoning:** Creates 3,5-dimethyl-4-methoxy pattern

**Environment State (t=2):**
Current Assembly: Benzimidazole-methoxy with sulfinylmethyl attached
Connected Components: 1
Remaining Motifs: [Motif_3, Motif_4]
Remaining Target Edges: 2
Current Properties: Basic scaffold formed, needs completion

**Motif 4 (Methoxy -methyl)**

**Analysis:** Should create branched substitution on pyridine
**Proposal:** Connect to Motif_3 - my carbon to C5, oxygen bridges to C4
**Chemical Impact:** Completes electron-rich pyridine
**Note:** This creates the specific substitution needed for activity

**Reward Calculation:**
r_valid = +1.0
r_stable = -0.1 (some steric crowding)
r_func = +0.9 (completed pyridine substitution)
r_conn = 0 (new component formed)
r_edge = +1.0 (matched target pattern)
**Total: R = 2.8**

**Central Arbitrator**

**Special Case:** Motif_4 bridges two positions on pyridine
**Creating:** 3,5-dimethyl-4-methoxy substitution pattern
**Decision: Execute complex bridging connection**

**Action Executed:** Connect: Motif_3[C5,C4] --bridge--> Motif_4[carbon,oxygen]

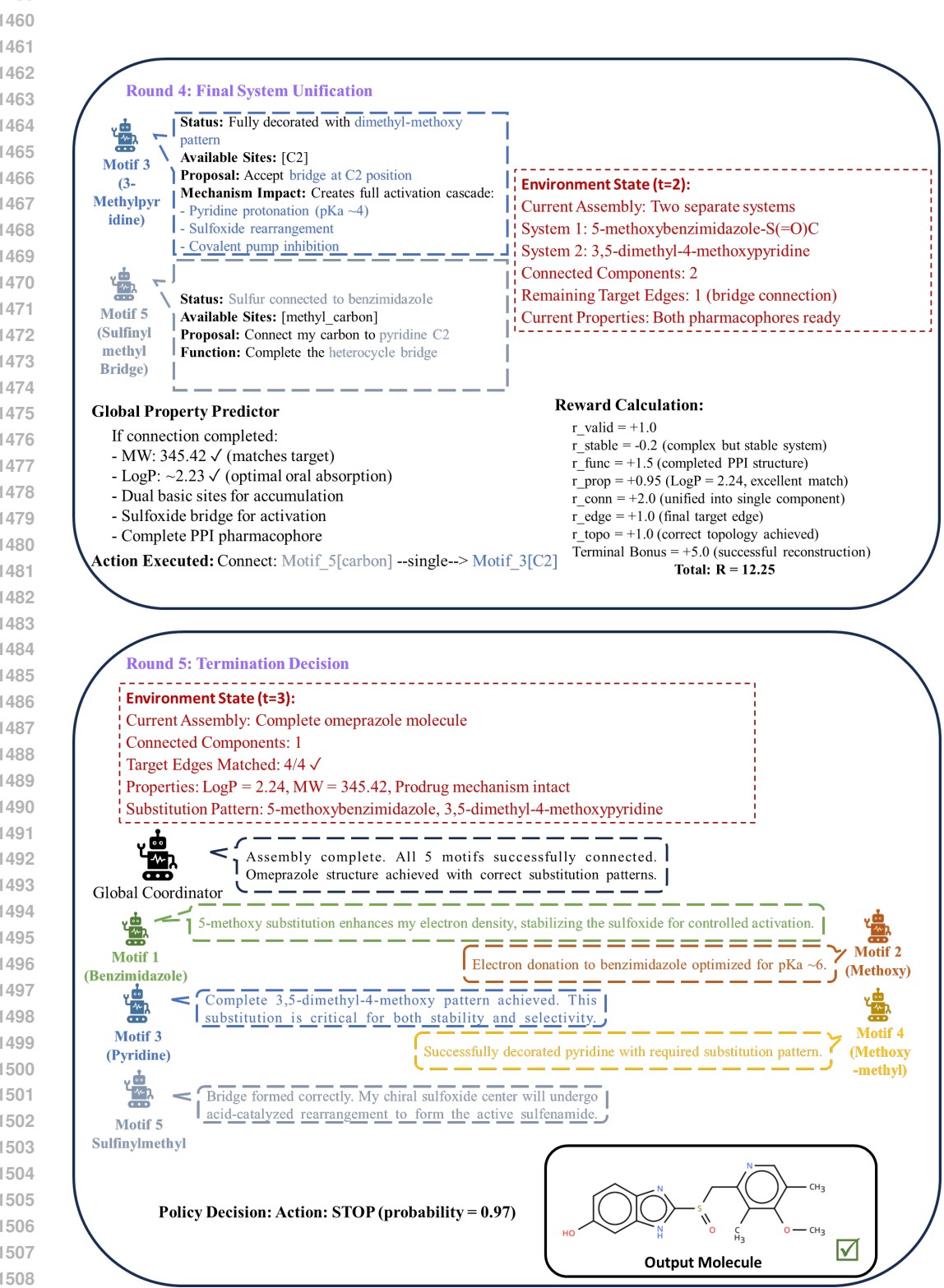

**Round 4: Final System Unification**

**Motif 3 (3-Methylpyridine)**

**Status:** Fully decorated with dimethyl-methoxy pattern
**Available Sites:** [C2]
**Proposal:** Accept bridge at C2 position
**Mechanism Impact:** Creates full activation cascade:
- Pyridine protonation (pKa ~4)
- Sulfoxide rearrangement
- Covalent pump inhibition

**Environment State (t=2):**
Current Assembly: Two separate systems
System 1: 5-methoxybenzimidazole-S(=O)C
System 2: 3,5-dimethyl-4-methoxypyridine
Connected Components: 2
Remaining Target Edges: 1 (bridge connection)
Current Properties: Both pharmacophores ready

**Motif 5 (Sulfinyl methyl Bridge)**

**Status:** Sulfur connected to benzimidazole
**Available Sites:** [methyl_carbon]
**Proposal:** Connect my carbon to pyridine C2
**Function:** Complete the heterocycle bridge

**Global Property Predictor**

If connection completed:
- MW: 345.42 ✓ (matches target)
- LogP: ~2.23 ✓ (optimal oral absorption)
- Dual basic sites for accumulation
- Sulfoxide bridge for activation
- Complete PPI pharmacophore

**Reward Calculation:**

r_valid = +1.0
r_stable = -0.2 (complex but stable system)
r_func = +1.5 (completed PPI structure)
r_prop = +0.95 (LogP = 2.24, excellent match)
r_conn = +2.0 (unified into single component)
r_edge = +1.0 (final target edge)
r_topo = +1.0 (correct topology achieved)
Terminal Bonus = +5.0 (successful reconstruction)
**Total: R = 12.25**

**Action Executed:** Connect: Motif_5[carbon] --single--> Motif_3[C2]

**Round 5: Termination Decision**

**Environment State (t=3):**
Current Assembly: Complete omeprazole molecule
Connected Components: 1
Target Edges Matched: 4/4 ✓
Properties: LogP = 2.24, MW = 345.42, Prodrug mechanism intact
Substitution Pattern: 5-methoxybenzimidazole, 3,5-dimethyl-4-methoxypyridine

Assembly complete. All 5 motifs successfully connected. Omeprazole structure achieved with correct substitution patterns.

Global Coordinator

Motif 1 (Benzimidazole)

5-methoxy substitution enhances my electron density, stabilizing the sulfoxide for controlled activation.

**Motif 2 (Methoxy)**

Electron donation to benzimidazole optimized for pKa ~6.

**Motif 3 (Pyridine)**

Complete 3,5-dimethyl-4-methoxy pattern achieved. This substitution is critical for both stability and selectivity.

**Motif 4 (Methoxy -methyl)**

Successfully decorated pyridine with required substitution pattern.

**Motif 5 Sulfinylmethyl**

Bridge formed correctly. My chiral sulfoxide center will undergo acid-catalyzed rearrangement to form the active sulfenamide.

**Policy Decision: Action: STOP (probability = 0.97)**

**Output Molecule**

**Episode Summary**

Step 1: Benzimidazole + methoxy (electron activation)
Step 2: Added sulfinylmethyl bridge to C2
Step 3: Decorated pyridine with methoxy-methyl fragment
Step 4: United systems via bridge completion
Step 5: Termination
**Total Reward:** 22.75 (across 4 connection steps)
Assembly Efficiency: 100% (all motifs correctly incorporated)
Property Achievement: LogP 2.24 vs 2.23 target (99.6% accuracy)
**Key Insights:**
- 5-methoxy on benzimidazole crucial for stability
- Sulfoxide bridge enables pH-dependent activation
- 3,5-dimethyl-4-methoxy pyridine pattern essential for selectivity
- Assembly order: core activation → bridge formation → decoration → unification

**Chemical Explanation Generated**: Omeprazole assembled through strategic 5-motif construction. Starting from benzimidazole core, we added 5-methoxy for electron activation, attached the sulfinylmethyl bridge at C2, separately decorated the pyridine with methoxy-methyl to achieve 3,5-dimethyl-4-methoxy pattern, then united both systems through the bridge. The assembly demonstrates how PPIs are built from heterocyclic cores with precise substitution patterns: benzimidazole for enzyme binding, sulfoxide for acid-activated prodrug mechanism, and electron-rich pyridine for selective accumulation in acidic compartments. The multi-step assembly preserves the critical structural features while building complexity systematically.

