# OpenReview forum: "MotifAgent: Motif-based Multi-Agent Graph-Language Alignment for Molecular Understanding and Generation"
_ICLR.cc/2026/Conference — ICLR 2026 Conference Withdrawn Submission_

### Official Review · Reviewer_Z277 · 2025-10-25

**Soundness:** 1
**Presentation:** 2
**Contribution:** 2
**Rating:** 2
**Confidence:** 5

**Summary:**

This paper introduces MotifAgent, a multi-agent reinforcement learning (MARL) framework that formulates molecular assembly as a collaborative decision-making problem. Each molecular motif is represented by an LLM-based agent that shares a common policy backbone and negotiates connections with other motifs to reconstruct 2D molecular topology. Experiments show improvements over existing LLM-based molecular models on molecular description generation, reaction prediction, and property prediction.

**Strengths:**

1. Explicitly define and model molecular assembly rules that typical LLMs overlook, addressing a real weakness in current molecule–text systems.
2. Novel design using inter-agent negotiation, Set-BC, and MAPPO + topology-aware reward shaping.
3. Experiments show gains over baselines.

**Weaknesses:**

## The motivation is not clear and needs evidence
1.1
> Abstract: "But existing approaches can only identify motifs without understanding their topological connection rules and assembly principles"
>
It is unclear what the "existing approaches" refer to and why they fail to understand molecular structures and rules.

1.2
> Abstract: "preventing models from grasping the generative mechanisms of molecules"
>
It is unclear what the term “generative mechanisms of molecules” refers to. I assume it means how chemists synthesize molecules in the lab, which involves a series of chemical reactions. In that case, although the paper evaluates the method on single-step reactions, there is no evidence showing that the method addresses retrosynthesis or advances this direction.

1.3
> Abstract: We formulate molecular assembly as a collaborative multi-agent problem, where each motif is represented by an agent sharing a common LLM backbone
>
Even though we can assume that problems 1.1–1.2 exist, there is no clear justification for why a “collaborative multi-agent” approach is needed to address them. Why not simply train a new molecular model or fine-tune existing LLMs?

1.4
> Intro: This linearization process (note: SMILES linearization) inherently destroys the connectivity information present in two-dimensional molecular topology
>
Is there any specific evidence supporting this claim? One can easily recover the molecular graph structure from its SMILES representation. What exact information does the SMILES representation destroy?

1.5
After reading the second paragraph (lines 47–65) in the introduction, it appears that motif-based models should be used rather than LLMs for molecular generation. The work also cites several related works [1,2]. However, first, these models are not included in the experimental comparison. Second, it remains unclear why agents (LLMs) are needed in this context instead of purely motif-based molecular models. Third, the work selectively compares models in molecular description generation rather than structure generation, which confuses the writing here.

## Many missing related works

The paper mainly focuses on LMs-based molecular design and needs to be complemented with related work in the domains of molecular generation, inverse molecular design, molecular optimization, and (multimodal) LLM-based molecular modeling.

## The evaluation is incomplete and not robust

3.1 There is a lack of evaluation on molecular structure generation tasks, whether based on structural description texts or property description texts.

3.2 In molecular caption generation, the evaluation metrics are not reliable. Metrics such as BLEU and ROUGE are mainly adapted from text generation, where they measure the overlap between a generated and a reference sentence. This makes sense for natural language but not for molecular texts, as these metrics cannot distinguish subtle yet critical differences, such as between an amine and an amide.

## Wrong reference

In line 359, the reference to ChEBI-20 is incorrect. Both ChEBI-20 and MoleculeSTM are from PubChem, and it is unclear how the training set differs from the test set.

## Reference:

[1] Molecule generation for target protein binding with structural motifs. ICLR 2023.

[2] De Novo Molecular Generation via Connection-aware Motif Mining. ICLR 2023.

**Questions:**

1. Efficiency: MARL with LLM backbones is computationally expensive; runtime and scalability are not reported.

2.  Interpretability: It is unclear how the model’s learned assembly decisions correspond to chemical intuition (e.g., can it explain why certain motifs connect?).

3. The method needs to be tested for the transfer performance to unseen motif vocabularies.

4. There is a lack of sensitivity analysis of critic fusion weights.

---

### Official Review · Reviewer_GtqH · 2025-10-31

**Soundness:** 3
**Presentation:** 2
**Contribution:** 3
**Rating:** 4
**Confidence:** 3

**Summary:**

This paper proposes MotifAgent, a motif-based multi-agent RL framework that aligns molecular graphs with language to both reconstruct molecules and generate new ones under chemical/topological constraints. Each motif is an “agent” sharing a common LLM backbone; agents propose connections (motif–site–bond type) through a hierarchical action space while a central arbitrator filters invalid proposals and scores valid ones. Training uses CTDE + MAPPO, Set-based Behavioral Cloning (Set-BC) to handle multiple topologically equivalent assembly paths, and topology-aware reward shaping (validity, connectivity, target-edge progress, graph-edit distance, etc.). Experiments report SOTA or competitive results on (i) ChEBI-20 description generation, (ii) Mol-Instructions reaction prediction (including retrosynthesis), and (iii) MoleculeNet classification, plus ablations showing multi-agent > single-agent and Set-BC > fixed-order supervision.

**Strengths:**

1. Conceptual novelty & clarity of formulation. Treating motifs as cooperating agents with CTDE + LLM policy is original and well-motivated; the hierarchical sampler (motif→site→target→bond) plus validity masks grounds actions in chemistry/topology.

2. Principled learning signals. Set-BC properly handles path multiplicity; reward shaping explicitly targets connectivity, target-edge coverage, and topological distance—beyond generic sequence likelihood.

3. Central arbitrator for global consistency. The two-phase screen + priority scoring formalizes action selection under chemistry/topology constraints.

4. Retrosynthesis (Mol-Instructions): higher exact match and fingerprint similarities; authors also claim 100% chemical validity for generated strings.

5. MoleculeNet classification: average ROC-AUC 77.19, surpassing several specialist/LLM baselines.

6. Ablations are informative. Multi-agent vs single-agent and Set-BC vs fixed-order show clear wins and better topology preservation / sample efficiency.

**Weaknesses:**

1. Reproducibility and config inconsistencies. Appendix A says 32 parallel envs with rollout 64 on 8×A100; Table 5 lists 8 envs with rollout 32 and different LRs—these conflict. Code is promised only upon acceptance. Please reconcile and provide exact reproduce-ready configs/seeds.

2. Evaluation clarity/fairness.

• MoleculeNet: splits (random vs scaffold), featurization, and instruction-conversion details are not described here—hard to compare to strong 2D/3D GNN baselines.

• 3D models (e.g., Uni-Mol) are cited but it’s unclear if directly compared on identical splits/metrics.

3. Description generation: table shows many specialist baselines, but protocol parity (tokenizers, decoding, length constraints) isn’t fully specified in the main text snippets.

4. “Chemical validity” definition is ambiguous. The method enforces valence/aromaticity/topological checks during environment steps, yet the reported 100% chemical validity in reaction tasks may reflect string/formal validity (e.g., SELFIES) rather than full RDKit sanitization + stereochemistry constraints; please define exactly what is measured and how.

5. Ablation surface is incomplete. No explicit ablations for the arbitrator (e.g., removing the scoring term or varying weights) or for individual topology-aware reward terms; this would isolate where the gains come from beyond multi-agent/Set-BC.

6. Compute and efficiency. Training on large LLM backbones (Qwen2.5-7B policy, MolT5 critic) with multi-env RL is expensive; wall-clock, tokens/step, and inference latencies (beam search k=5) are not reported, which is important for practical adoption.

**Questions:**

1. Splits & protocols. What splits were used on MoleculeNet (scaffold vs random)? Are the reported numbers directly comparable to prior SOTA graph/3D models under identical splits and evaluation scripts?

2. Chemical validity metric. In retrosynthesis, does “100% chemical validity” mean successful SELFIES decoding, RDKit sanitization, or additional stereochemistry/valence checks? Please specify the exact pipeline.

3. Arbitrator ablation. What happens if the arbitrator’s scoring (Eq. 2) is removed or weights $w_1, w_2, w_3$ are varied? Any sensitivity analysis?

4. Reward shaping sensitivity. How critical are $r_{conn}, r_{edge}, r_{topo}$? Please show a topology-shaping ablation.


5. Compute and throughput. Please reconcile the conflicting training configs (envs/rollout/learning rates) and report wall-clock time, tokens processed, and inference latency (per molecule) on each task.


6. OOD generalization. How does MotifAgent perform when test molecules contain unseen motifs or motif-connection rules? Any analysis of failure modes under OOD motif sets?

7. 3D awareness. Since many properties hinge on 3D conformation, can the critic or rewards incorporate coarse 3D signals (e.g., distance constraints) without heavy QM? Any plans for this?

---

### Official Review · Reviewer_6duw · 2025-11-01

**Soundness:** 3
**Presentation:** 2
**Contribution:** 3
**Rating:** 6
**Confidence:** 2

**Summary:**

This paper introduces MotifAgent, a novel collaborative multi-agent reinforcement learning framework designed to reformulate the molecular assembly task as a sequence of interactions between heterogeneous language model (LM) agents. At each step, the agents select two molecular fragments, connection sites, and bond types to connect two disjoint fragments.

As a complementary contribution, the authors propose a topology-aware reward function combined with a multi-agent proximal policy optimization (MAPPO) strategy. Experimental results demonstrate significant improvements over both specialist and generalist baselines across three task categories: molecular description, molecular property prediction, and reaction prediction.

**Strengths:**

The idea of framing molecular assembly as a sequence of LLM-agent interactions trained via reinforcement learning is moderately novel and represents a creative extension of traditional fragment-based molecular generation.

The experimental results and ablation studies appear comprehensive and support the claimed advantages over baseline methods.


The paper is clearly written and well-structured, with intuitive illustrations that effectively convey the main ideas.

**Weaknesses:**

Although the MotifAgent framework is described primarily in the context of molecular assembly, the experiments are conducted on diverse tasks such as molecular description, property prediction, and reaction prediction. It remains unclear, whether the RL-based molecular assembly phase was trained before, after, or in conjunction with downstream task training. Does the molecular assembly process interacts with or influences the reasoning steps for downstream tasks — for example, whether molecule assembly occurs implicitly during the “thinking process” before final prediction generation?

Clarifying this setup—perhaps through a detailed training pipeline diagram or a step-by-step experimental procedure in the Supplementary Materials—would greatly improve the paper’s readability and the community’s ability to reproduce and extend the work.

**Questions:**

After reading the paper, my only remaining question concerns the adaptation of MotifAgent to downstream tasks, as outlined in Weaknesses. Specifically, how is the model fine-tuned or conditioned for different task types, and does the MotifAgent remain active (or frozen) during those downstream evaluations? Clarifying this would help assess the framework’s flexibility and generalization potential.

---

### Official Review · Reviewer_tQYM · 2025-11-01

**Soundness:** 1
**Presentation:** 2
**Contribution:** 2
**Rating:** 2
**Confidence:** 4

**Summary:**

This paper introduces MotifAgent, a multi-agent reinforcement learning framework designed to learn the generative principles of molecular assembly. MotifAgent fragments a molecule into motifs, treating each motif as an independent "agent" that shares a common LLM backbone. These agents then negotiate and propose connections to reconstruct the molecule's 2D structure, guided by a centralized critic and arbitrator within a Centralized Training, Decentralized Execution framework. The model uses Set-based Behavioral Cloning to handle ambiguous assembly orders and topology-aware rewards to ensure chemical validity.

**Strengths:**

1. Applying LLMs to help the molecule understanding is crucial.
2. Building a multi-agent system to study molecule assembly is interesting.

**Weaknesses:**

1. My main concern is experiments. The paper compares MotifAgent against a wide range of "Specialist Models" and "LLM-Based Generalist Models" (e.g., in Table 1). However, it does not explicitly state that these baselines were retrained from scratch using the exact same dataset (the 51,340 pairs from MoleculeSTM)  and the same backbone model. This makes it difficult to "isolate" the true source of the performance gain. We can't be certain if MotifAgent's superior performance comes purely from its novel multi-agent framework or if it's also influenced by a stronger backbone model, a different or higher-quality training dataset than the ones used by the baseline models.
2. State-of-the-art reasoning studies increasingly report results across diverse model families, parameter scales, and datasets to show that gains are not backbone- or data-specific. This paper focuses largely on a single backbone–dataset combination, making it hard to separate true algorithmic improvements from the chosen model or dataset. Additional experiments would substantially strengthen the claims.
3. The experimental validation is incomplete. Ablations cover only two coarse choices (multi-agent vs. single-agent and Set-BC vs. fixed-order) and omit critical components. In particular, there is no test of the four-phase curriculum to show that staged training beats training from scratch. Nor are the effects of the arbitrator’s scoring, the critic’s auxiliary heads, or the many (8+) reward terms disentangled, so it’s unclear which parts actually drive the gains.

**Questions:**

Please refer to the weaknesses.

---

### Note · Authors · 2025-12-01

I have read and agree with the venue's withdrawal policy on behalf of myself and my co-authors.